# Reining Generalization in Offline Reinforcement Learning via Representation Distinction

**Yi Ma**
College of Intelligence and Computing
Tianjin University
mayi@tju.edu.cn

**Hongyao Tang**[*]
Université de Montréal
Mila
tang.hongyao@mila.quebec

**Dong Li**
Noah's Ark Lab, Huawei Technology
dongleecsu@gmail.com

**Zhaopeng Meng**
College of Intelligence and Computing
Tianjin University
mengzp@tju.edu.cn

## Abstract

Offline Reinforcement Learning (RL) aims to address the challenge of distribution shift between the dataset and the learned policy, where the value of out-of-distribution (OOD) data may be erroneously estimated due to overgeneralization. It has been observed that a considerable portion of the benefits derived from the conservative terms designed by existing offline RL approaches originates from their impact on the learned representation. This observation prompts us to scrutinize the learning dynamics of offline RL, formalize the process of generalization, and delve into the prevalent overgeneralization issue in offline RL. We then investigate the potential to rein the generalization from the representation perspective to enhance offline RL. Finally, we present Representation Distinction (RD), an innovative plug-in method for improving offline RL algorithm performance by explicitly differentiating between the representations of in-sample and OOD state-action pairs generated by the learning policy. Considering scenarios in which the learning policy mirrors the behavioral policy and similar samples may be erroneously distinguished, we suggest a dynamic adjustment mechanism for RD based on an OOD data generator to prevent data representation collapse and further enhance policy performance. We demonstrate the efficacy of our approach by applying RD to designed backbone algorithms and widely-used offline RL algorithms. The proposed RD method significantly improves their performance across various continuous control tasks on D4RL datasets, surpassing several state-of-the-art offline RL algorithms.

## 1 Introduction

Reinforcement learning (RL) is a machine learning paradigm centered on training intelligent agents to make decisions through interactions with an environment. In this learning process, an agent develops a policy—a mapping from states to actions—via trial and error, with the goal of maximizing cumulative rewards over time. However, acquiring interaction data in many real-world applications can be expensive, time-consuming, or even hazardous. This spurred the advancement of offline Reinforcement Learning, which has garnered considerable interest in the research community due to its ability to learn effective policies from pre-existing data collected by unknown behavioral policy

---

[*]Corresponding author.

37th Conference on Neural Information Processing Systems (NeurIPS 2023).

without necessitating online interactions with the environment. Offline RL can result in substantial savings in resources, time, and the risk associated with online exploration.

Despite its appealing potential, offline RL encounters challenges in learning optimal policies from limited and suboptimal datasets. The primary challenge in offline RL is the distribution shift between the dataset and the learned policy. As the agent updates its policy, it may come across out-of-distribution (OOD) state-action pairs absent from the limited support of the fixed offline dataset [1]. Conventional RL algorithms, such as SAC [2] or TD3 [3], could yield overly optimistic Q-value estimations for these unseen state-action pairs due to overgeneralization, leading to catastrophic performance. Existing methods in offline RL seek to address this challenge by incorporating techniques like policy constraint [4, 5, 6, 7], conservative value estimates, [8, 9], and uncertainty estimation [10, 11, 12].

Recently, [13] elucidated that the ability of existing offline reinforcement learning (RL) methods to handle OOD actions is significantly attributed to their impact on the learned representations of state-action pairs. The learned representation can capture the underlying structure and essential aspects of the state-action space, enabling the agent to identify similarities and patterns across various state-action pairs, thus improving agent's learning efficacy and generalization capacity. Although there are a few works [14] also study representation learning in offline RL, they focus on how different kinds of pretrained representation matter in downstream policy learning, neglecting the understanding of the essential interplay between representation and policy during the offline co-learning process.

In this paper, we first present a view dubbed Backup-Generalization Cycle to gain a tangible comprehension of how a typical offline RL algorithm learns. We subsequently formalize how generalization happens, and discuss the overgeneralization issue prevalent in offline RL. Finally, we introduce a novel method called Representation Distinction (RD) to mitigate overgeneralization from in-sample state-action pairs to OOD ones. Specifically, we distinctly differentiate the data sourced from the dataset and the data generated by the learning policy from the representation perspective. In cases where the learning policy's performance aligns with the behavioral policy and the generated data largely overlaps with the dataset, we devise an OOD data generator to produce data with lower Q-values than the current policy. We then gradually shift our focus from differentiating in-sample data and data generated by the learning policy to differentiating data generated by the learning policy and data generated by the OOD actor in the representation space, preventing similar samples from being erroneously distinguished.

To demonstrate the efficacy of our proposed RD method in enhancing offline RL policy performance by reining the generalization, we apply it to two backbone algorithms, which are modified versions of the original SAC and TD3 algorithms by incorporating an ensemble of critics and an uncertainty regularization term in the policy update process. By integrating RD into these simple algorithms, our proposed method RD can improve the performance of the backbone agents on D4RL datasets across various continuous control tasks and outperforms several state-of-the-art offline RL algorithms. We also apply RD on two widely-used baselines TD3BC and CQL and improve their performance significantly. In-depth analysis based on visualization and statistical results that demonstrates the efficacy of RD in obtaining more differentiable representation compared to its variants is also provided.

The main contributions of this work can be summarized as follows:

- We introduce a view called Backup-Generalization Cycle to foster an understanding of typical offline value function learning, and highlights the necessity of reining generalization to enhance offline RL.
- We proposed a practical plug-in method Representation Distinction (RD) from the perspective of representation to enhance the performance of offline RL methods by inhibiting overgeneralization among state-action pairs sourced from different distributions.
- We evaluate the effectiveness of RD by applying it to designed backbone algorithms and existing widely-used algorithms and enhance their performance significantly.

## 2   Related Works

**Offline RL**   Offline RL algorithms strive to train RL agents using pre-collected datasets. Nevertheless, the distribution shift between the behavior policy and the policy being learned may lead to issues, as OOD actions are sampled from the learned policy and incorporated into the learned

critic. To address this challenge, various approaches have been proposed. Some earlier methods aim to constrain the learned policy to remain close to the behavior policy, which can be accomplished through explicit policy regularization [4, 5, 6], implicit policy constraints [15, 16, 17, 18, 19], or by employing auxiliary behavioral cloning losses [7]. Alternative approaches penalize the Q-value of OOD actions to discourage their selection [8, 9, 10, 11, 12]. Moreover, model-based methods that train with conservative penalties have been suggested [20, 21, 22, 23, 24, 25].

**Representation in Offline RL.**   Prior research has aimed to analyze various aspects of the representations induced by TD-based methods using function approximation, predominantly in the standard online RL setting [26, 27, 28, 29, 30]. More recently, this line of inquiry has emerged in the offline RL setting [31, 32, 33, 34, 13]. [31] investigates which representations can result in stable convergence of TD in a linear setting. [32, 33] explore the learning dynamics of Q-learning in an overparameterized setting, observing excessively low-rank and aliased feature representations at the fixed points identified by TD-learning. [34] studies the extent of impact of different interventions on the causal link between the effective rank and offline RL agent performance and points out that there is no strong relationship between them. [13] highlights that a substantial portion of the benefits of existing offline RL approaches, which aim to avoid OOD actions, actually originates from their effects on the learned representations. The authors also identify specific metrics that facilitate effective evaluation of the quality of the learned representation. [14] pretrain the representation using different auxiliary loss and then fix the representation to apply it to downstream policy learning. Nevertheless, a thorough examination of the explicit representation distinction between in-sample and OOD state-action pairs in the context of offline RL is lacking. Our approach promotes orthogonality between the representation vectors of in-sample and OOD data, offering a more practical solution for inhibiting overgeneralization, thus enhancing offline RL.

# 3   Preliminary

Markov Decision Process (MDP) is a mathematical framework that models decision-making processes in stochastic environments. It is defined by a tuple $(S, A, P, R, \gamma)$, where $S$ is the set of states, $A$ is the set of actions, $P$ is the state transition probability function, $R$ is the reward function, and $\gamma$ is the discount factor. Q-learning is the major approach for obtaining the optimal $\pi(s)$, which learns a Q-value function $Q(s, a)$ that represents the expected cumulative discounted rewards when starting from the state $s$ taking the action $a$ and executing the policy $\pi$ thereafter. the Q-value function is evaluated by iterating the Bellman operator as $\mathcal{T}Q(s, a) = \mathbb{E}_{s' \sim P(\cdot|s,a)} \left[ r(s, a) + \gamma \mathbb{E}_{a' \sim \pi(s')} Q(s', a') \right]$. The goal of RL is to find an optimal policy $\pi(s)$ that maximizes the cumulative discounted rewards.

Offline Reinforcement Learning [1], aims to learn a policy from a fixed dataset of interaction samples, without further interaction with the environment. The dataset $\mathcal{D}$ consists of transition tuples $(s, a, r, s')$ collected from interactions between one or multiple behavior policies and the environment. The goal is to learn an optimal policy using only this dataset.

To deal with large state-action space, deep RL resorts to deep neural networks for function approximation. Typically, a Q-network parameterized by $\phi$ can be viewed as $Q_\phi(s, a) = \mathbf{w}^\top \Phi(s, a)$, where $\Phi(s, a) \in \mathbb{R}^d$ is regarded as the representation obtained by the penultimate layer of Q-network while $\mathbf{w} \in \mathbb{R}^d$ is the linear weight [35, 36, 33, 13]. In conventional online and offline RL learning process, the representation and the linear weight learn together. A few recent works[37, 38, 39, 40] also introduce auxiliary objectives to strengthen or regularize the representation with different purposes. We use this view for the presentation of our analysis and proposed method in the following sections.

# 4   Reining Generalization in Offline RL

In this section, we focus on the learning dynamics of offline RL. First, we present a view called Backup-Generalization Cycle (Sec. 4.1) to gain a tangible comprehension of how a typical offline RL algorithm learns. We then take a further step to formalize how generalization happens, and discuss the overgeneralization issue in offline RL (Sec. 4.2). Finally, we propose the idea of kernel control to address the overgeneralization issue (Sec. 4.3), with the practical implementation deferred to Sec. 5.

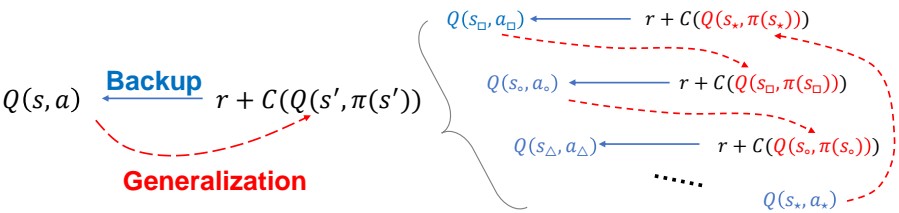

Figure 1: An illustration of Backup-Generalization Cycle. $s_\square, s_\circ, s_\triangle, s_\star$ denotes four states in the offline dataset $\mathcal{D}$. Backups and generalizations are denoted by blue and red arrows respectively.

## 4.1 Backup-Generalization Cycle in Offline RL

Offline RL learns from a static offline dataset with no access to online interaction over the course of policy learning. To gain a better understanding of the intricacies of offline deep RL, for the first, we introduce a view called Backup-Generalization Cycle. This view, as depicted in Fig. 1, fosters an understanding of typical offline value function learning via two key components: **Backup** and **Generalization**:

- Typical offline deep RL algorithms employ various forms of **backups** to learn the $Q$-function network. Here we use $C(\cdot)$ to denote a generic form of the target value such as Clipped Double $Q$-Learning [3]. It is notable that only the values of state-action pairs in the offline dataset are *directly* updated.

- Since the values of all state-action pairs that are not in the dataset will never be directly updated, changes to the values of such $(s, a) \notin \mathcal{D}$ are solely *indirectly* instigated by the backups on $(s, a) \in \mathcal{D}$, i.e., through **generalization**. Note that the state-action pairs for the target values of backup are highly likely to be absent from the dataset, i.e., $(s', \pi(s')) \notin \mathcal{D}$, as the learning of current policy $\pi$.

Hence, the typical offline learning process of value function can be regarded as a consequence of the complex interplay between backup and generalization. This dynamic interplay forms a cycle: (1) the backups on $(s, a) \in \mathcal{D}$ consistently influence the values of $(s, a) \notin \mathcal{D}$ (highly likely including $(s', \pi(s'))$); (2) the consistently changing $Q(s', \pi(s'))$ participates in the backups on $(s, a) \notin \mathcal{D}$; the two kinds of dynamics iterate and twine during the learning process.

There are also some similar opinions mentioned in a few recent works on offline RL [41, 29]. We utilize the Backup-Generalization Cycle to distinctly separate and better analyze these two contributing factors. In the next subsection, we concentrate on the overgeneralization issue with a formal characterization of value generalization among state-action pairs.

## 4.2 Overgeneralization in Offline RL

According to the Backup-Generalization Cycle presented above, generalization plays a significant role for $(s, a) \notin \mathcal{D}$. In contrast to backup, which is relatively explicit and controllable, generalization is implicit and intricate. In the following of this subsection, we analyze value generalization among state-action pairs during the offline RL learning process.

For a starting case, we consider how the $Q$ function update caused by typical Temporal-Difference (TD) learning on a single state-action pair $(s, a) \in \mathcal{D}$ (denoted as $\phi \to \phi'$), affects the $Q$-value of an arbitrary state-action pair $(\bar{s}, \bar{a})$. The post-update parameter $\phi'$ can be formalized as follows:

$$\phi' = \phi + (\mathcal{T}Q_\phi(s, a) - Q_\phi(s, a)) \nabla_\phi Q_\phi(s, a), \tag{1}$$

where learning rate is omitted for convenience. We further formalize the post-update $Q$-value of $(\bar{s}, \bar{a})$ by Taylor expansion at the pre-update parameter $\phi$:

$$Q_{\phi'}(\bar{s}, \bar{a}) = Q_\phi(\bar{s}, \bar{a}) + \nabla_\phi Q_\phi(\bar{s}, \bar{a})^\top (\phi' - \phi) + \mathcal{O}\left(\|\phi' - \phi\|^2\right), \tag{2}$$

Now, we can characterize the generalization from $(s, a)$ to $(\bar{s}, \bar{a})$ by plugging Eq.1 into Eq.2:

$$Q_{\phi'}(\bar{s}, \bar{a}) = Q_\phi(\bar{s}, \bar{a}) + k_\phi(\bar{s}, \bar{a}, s, a)(\mathcal{T}Q_\phi(s, a) - Q_\phi(s, a)) + \mathcal{O}\left(\|\phi' - \phi\|^2\right) \tag{3}$$

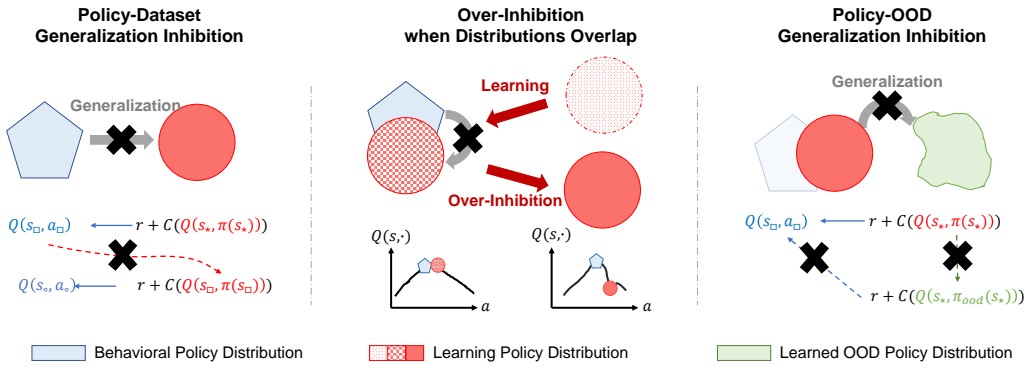

Figure 2: A conceptual illustration of Reining Generalization via Kernel Control.

where $k_\phi(\bar{s}, \bar{a}, s, a) \doteq \nabla_\phi Q_\phi(\bar{s}, \bar{a})^\top \nabla_\phi Q_\phi(s, a)$ and it is called Neural Tangent Kernel in [27]. Eq.3 elucidates the change of the $Q$-value of any state-action pair $(\bar{s}, \bar{a})$ can be mainly characterized by the kernel $k_\phi(\bar{s}, \bar{a}, s, a)$ and the TD-error $\mathcal{T}Q_\phi(s, a) - Q_\phi(s, a)$. Apparently, the kernel controls the extent of generalization: for small values of $k_\phi(\bar{s}, \bar{a}, s, a)$, the generalization effect is minor and the classic tabular learning (i.e., no generalization) can be viewed as a special case when $k_\phi$ and the higher-order term are 0; for large values of $k_\phi(\bar{s}, \bar{a}, s, a)$, the generalization effect is prominent and $Q_{\phi'}(\bar{s}, \bar{a})$ can be largely changed in either the same or the opposite direction of TD-error. Intriguingly, Eq. 3 indicates that we can control the generalization by mainly adjusting the kernel $k_\phi(\bar{s}, \bar{a}, s, a)$.

It is widely deemed that the crux of offline RL is the overestimation of OOD data, i.e., $(s, a) \notin \mathcal{D}$. Recall the Backup-Generalization Cycle in Fig. 1. The overestimation propagates directly via backups and further spreads via generalization. Most existing works on offline RL focus on eliminating or suppressing overestimation by conservative value estimation [8, 10] or distribution constraint [4, 5], leading to various forms of $C(\cdot)$. These works can be categorized as overestimation-control backups. However, since the source of overestimation is improper generalization, i.e., **overgeneralization** issue, we argue that it is inadequate to only cut the *downstream* while leave the *upstream* uncontrolled. Naturally, the generalization should also be carefully controlled as a preposed complement to the overestimation-control backups. Therefore, we take a further step and propose a novel method to control the generalization in the next subsection.

### 4.3 Reining Generalization via Kernel Control

Since the control of generalization (i.e., the *upstream*) is neglected in most previous offline RL works, we propose a novel method to address the overgeneralization issue by controlling the kernel in Eq. 3. The key idea is a two-stage generalization control: Policy-Dataset Generalization Inhibition and Policy-OOD Generalization Inhibition. Fig. 2 illustrates the concept of the two stages and an intermediate scene between them, which are detailed below.

As illustrated in the left of Fig. 1, the first stage of generalization control is to inhibit the generalization from behavior policy distribution (or dataset) to learning policy distribution, i.e., suppress $Q(s, a) \xrightarrow{\text{Generalization}} Q(s, \pi(s))$ for $(s, a) \in \mathcal{D}$. Such kind of generalization is called *extrapolation* in the offline RL literature [1]. Extrapolation is a main component of generalization as its consequence $Q(s, \pi(s))$ consistently participates in the backups. Thus, overestimation caused by improper extrapolation severely handicaps the learning. For a stable and effective learning process, we argue that extrapolation should be inhibited, especially at the early learning stage, where the learning policy is usually largely different with the behavior policy. This can be achieved by encouraging the kernel to 0, i.e., $\min_\phi |\nabla_\phi Q_\phi(s, a)^\top \nabla_\phi Q_\phi(s, \pi(s))|$. Consequently, training on $(s, a) \in \mathcal{D}$ induces only a minor change to $Q_\phi(s, \pi(s))$.

With effective suppression of overestimation achieved by the first stage of generalization inhibition, we further consider an intermediate scene where the learning policy evolves and resembles the behavioral policy as the training proceeds, leading to a distribution overlap. In this scene, the effect of the first stage could become over-inhibition. In another word, $\min_\phi |\nabla_\phi Q_\phi(s, a)^\top \nabla_\phi Q_\phi(s, \pi(s))|$ for $a \approx \pi(s)$ could lead to a sharp and bumpy $Q$-value landscape as shown in the middle of Fig. 2. It adversely impacts the robustness of $Q$ function, further resulting in a brittle policy. Hence, it's unwise

to impose the Policy-Dataset Generalization Inhibition all along the learning process. For some empirical evidence of this, we observe performance decay in some of our experiments in Table 3 and Fig.4. A natural remedy is to gradually remove the inhibition. However, it leaves value generalization uncontrolled once again, exposing the policy learning under the risk of collapse due to overestimation.

To address the dilemma above, we introduce the second stage, Policy-OOD Generalization Inhibition, as illustrated in the right of Fig. 2. In this stage, we make use of an additional policy $\pi_{\text{ood}}$ that outputs *OOD actions*. The OOD action here is defined as an action with a lower $Q$-value than the action selected by the learning policy for the same state. The actual effect of the second stage is suppression of the generalization between the learning policy distribution and OOD policy distribution. Such a means of kernel control can be viewed as a special way of increasing the action gap [42]. It improves the robustness of value function learning by discouraging erroneous generalization on suboptimal actions. Formally, this is achieved by $\min_\phi |\nabla_\phi Q_\phi(s, \pi(s))^\top \nabla_\phi Q_\phi(s, \pi_{\text{ood}}(s))|$.

Overall, our kernel control method consists of the two stages of generalization inhibition, which effectively controls the generalization over the entire course of learning, providing a favorable *upstream* for better offline RL.

## 5 Representation Distinction in Offline RL

In this section, we devise a practical algorithm to rein the generalization in offline RL following the design of two stages of generalization inhibition. A heuristic method to achieve smooth transition among them is also proposed. We term our algorithm of reining generalization as Representation Distinction (RD) and apply them to both existing and designed backbone algorithms.

### 5.1 Practical Algorithm Design

We first focus on the phase depicted in the left of Fig.2. A natural approach is to directly minimize $\mathbb{E}_{s,a\sim\mathcal{D}} |k_\phi(s, \pi(s), s, a)|$ across the dataset. However, due to the high dimensionality of $\nabla_\phi Q_\phi(s, a)$, the direct computation of $\mathbb{E}_{s,a\sim\mathcal{D}} |k_\phi(s, \pi(s), s, a)|$ involves the computation and backpropagation through per-example gradient dot products, which is computationally prohibitive. To mitigate this, we adopt the method proposed by [33] and approximate $\Delta(\phi)$ with the contribution solely from the last layer parameters, i.e., $\mathbb{E}_{s,a\sim\mathcal{D}} |\nabla_{\mathbf{w}} Q_\phi(s, \pi(s))^\top \nabla_{\mathbf{w}} Q_\phi(s, a)|$. Consequently, we derive the following loss function:

$$\mathcal{L}_1 = \mathbb{E}_{s,a\sim\mathcal{D}} |\Phi(s, \pi(s))^\top \Phi(s, a)| \tag{4}$$

where $\Phi(s, a)$ signifies the representation of state-action pairs as mentioned in Section 3. By minimizing $\mathcal{L}_1$, we encourage the learned Q-function to yield representations that are as orthogonal as possible between data from dataset and $\pi$. In this manner, the representations of $(s, a)$ and $(s, \pi(s))$ can be distinctly discerned.

To substantiate the efficacy of inhibiting generalization between in-sample data and data derived from the learning policy $\pi$, we conduct rudimentary experiments by directly minimizing $\mathcal{L}_1$ during the training of the standard off-policy SAC algorithm [2] on both halfcheetah-medium and halfcheetah-medium-expert. The evaluation results depicted in Fig.3 demonstrate that enhancing the representation alone, devoid

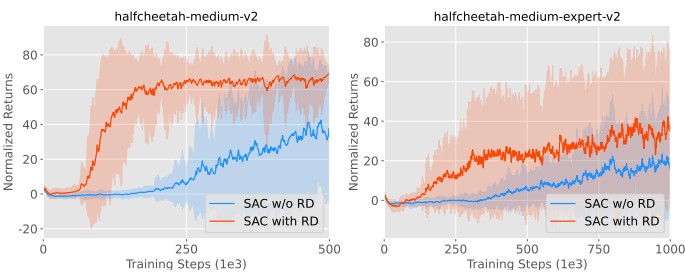

Figure 3: Average results of SAC over 5 seeds with and w/o RD.

of explicit pessimism, can achieve commendable performance, substantially surpassing the baseline SAC. These outcomes underline the potency of inhibiting generalization to eliminate overestimation in offline setting.

Further, we consider the phase in the right of Fig.2. To improve the robustness of the Q function by circumventing the excessive generalization of the Q-function to OOD data, we introduce a simple OOD data generator that creates data with lower Q-values compared to the data produced by current policy $\pi$. Specifically, the OOD data generator $\pi_{\text{ood}}$ is trained to generate OOD state-action pairs that satisfy the following condition: $\alpha * Q_\phi(s, a) < Q_\phi(s, \pi_{\text{ood}}(s)) < \beta * Q_\phi(s, a)$, where $Q_\phi(s, a)$ and $Q_\phi(s, \pi_{\text{ood}}(s))$ are assumed to be positive for ease of presentation, and $0 < \alpha < \beta < 1$ is the predefined hyperparameters. In practice, the optimization of $\pi_{\text{ood}}$ is implemented via the following margin ranking loss:

$$\mathcal{L}_{\text{OOD}} = \max\left(0, Q_\phi(s, \pi_{\text{ood}}(s)) - \beta * Q_\phi(s, a)\right) + \max\left(0, \alpha * Q_\phi(s, a) - Q_\phi(s, \pi_{\text{ood}}(s))\right) \quad (5)$$

As the offline RL training progresses, we gradually shift our focus from distinguishing in-sample data and data generated by $\pi$ to distinguishing data generated by $\pi$ and data generated by the OOD actor $\pi_{\text{ood}}$ in the representation space, i.e., $\mathcal{L}_2 = \underset{s \sim \mathcal{D}}{\mathbb{E}} |\Phi(s, \pi_{\text{ood}}(s))^\top \Phi(s, \pi(s))|$. In practice, a simple heuristic approach is proposed to help balance $\mathcal{L}_1$ and $\mathcal{L}_2$:

$$\mathcal{L}_{\text{RD}} = (1 - w) * \mathcal{L}_1 + w * \mathcal{L}_2 \quad (6)$$

where $w$ gradually increases according to $w = \tanh(\frac{\text{step\_num}}{\text{M}})$, step_num refers to the current training step, and M is a hyperparameter. During the update of $\pi_{\text{ood}}$, the gradient towards $Q_\phi$ is detached.

We would like to emphasize that the OOD generator is not required to be meticulously trained, as we only expect the Q value of the generated OOD data to be appropriately lower than that of the data generated by the current policy. This is due to the fact that the OOD data is not employed to directly influence the policy's decision-making process. Instead, it serves to inhibit the excessive generalization of the Q function, thereby enhancing the robustness.

## 5.2 Applying RD in Offline RL Algorithms

---

**Algorithm 1:** Offline RL with RD (Pseudocode in a PyTorch-like style)

---

```
1  # replay_buffer:  save dataset
2  # agent:  offline RL agent
3  # itv:  OOD actor and RD updating interval
4
5  # Loading dataset into replay buffer (omitted)
6  ...
7  for t in range(max_learning_steps):
8    agent.sample_data(replay_buffer)
9    # Update OOD actor and RD according to Eq.5 and 6 respectively at intervals
10   If t % itv == 0:
11     agent.update_ood_actor()
12     agent.update_RD()
13   # Update critic and actor following the offline RL algorithm design
14   agent.update_critic()
15   agent.update_actor()
16   ...
17 ...
```

---

While the introducing of RD to the offline learning process offers benefits, it alone cannot boost the performance of a standard online RL algorithm to a cutting-edge level, as depicted in Fig.3. This is because merely reining generalization from the representation perspective does not necessarily prevent overestimation of the Q-value. Our objective is to construct RD as a universally applicable plug-in to enhance any offline RL algorithm. Thus, to evaluate RD's versatility as a plug-in, we apply it in both existing algorithms and two designed rudimentary backbone algorithms named SAC-N-Unc and TD3-N-Unc. By modifying the original SAC/TD3 algorithm to employ a critic ensemble of number N and incorporate an uncertainty regularization term within the policy update process, we derive these backbone algorithms. It should be noted that these algorithms are not the original contribution of this paper as many similar uncertainty-based methods [10, 43, 12, 44, 45, 46] have been proposed. Details of the backbone algorithms are provided in the appendix.

Note that RD is algorithm-agnostic, meaning it can be applied to a wide range of actor-critic offline RL algorithms. Incorporating RD into offline RL algorithms is also straightforward, requiring only

the addition of two steps to the conventional policy updating scheme, as illustrated in Algorithm 1 in the appendix. In each training iteration, the OOD data generator is first trained to generate data with lower Q-values than the current policy. Then, the Q-function is updated by minimizing the loss function $\mathcal{L} = \epsilon * \mathcal{L}_{\text{RD}} + \mathcal{L}_{\text{critic}}$, and the policy is updated based on $\mathcal{L}_{\text{actor}}$. Here, $\epsilon$ represents a weighted factor, and $\mathcal{L}_{\text{critic}}$ and $\mathcal{L}_{\text{actor}}$ refer to the optimization objectives of any offline RL algorithm. A pseudocode using RD in offline RL is given in Algorothm 1. In the experimental section, we demonstrate the versatility of RD by integrating it into both backbone algorithms and existing offline RL algorithms.

# 6  Experiments

We evaluate RD by applying it on TD3BC [7], CQL [8], SAC-N-Unc and TD3-N-Unc through a series of experiments on D4RL [47] gym MuJoCo-v2 and Adroit-v1 datasets, where the former is a dataset commonly used in previous work for continuous control tasks, and the latter poses a significant challenge for most offline RL methods due to its sparse reward property. We compare our method with BC and several model-free offline RL algorithms, including DT [48], TD3BC [7], CQL [8], IQL [49], EDAC [10], and Diffusion-QL [50]. We obtain the results of the baselines by re-running the official codes or directly extracting them from the original papers. In our experiments, all the re-run baselines and our algorithm are executed with five random seeds, and we report the average normalized results of the final ten evaluations. We report the performance at 1M gradient step for TD3BC and CQL and that at 3M gradient step for SAC-N-Unc and TD3-N-Unc on MuJoCo tasks. For Adroit tasks, we report results at 500K gradient step. Note that due to the space limitation, we abbreviate the names of the datasets from {RANDOM, MEDIUM, MEDIUM-REPLAY, MEDIUM-EXPERT, EXPERT} to {R, M, MR, ME, E} in all the tables.

## 6.1  Main Results

Table 1: Results of different algorithms and the ones equipped with RD

| DATASET | TD3-N-UNC | TD3-N-UNC +RD | SAC-N-UNC | SAC-N-UNC +RD | TD3BC | TD3BC +RD | CQL | CQL +RD |
|---|---|---|---|---|---|---|---|---|
| HALFCHEETAH-M | $66.8 \pm 0.5$ | $\mathbf{66.8 \pm 1.2}$ | $65.9 \pm 1.0$ | $\mathbf{65.9 \pm 1.9}$ | $48.0 \pm 0.3$ | $\mathbf{48.3 \pm 0.5}$ | $47.1 \pm 0.2$ | $\mathbf{53.0 \pm 0.5}$ |
| HALFCHEETAH-MR | $53.4 \pm 3.9$ | $\mathbf{57.7 \pm 0.9}$ | $53.2 \pm 5.4$ | $\mathbf{61.5 \pm 1.4}$ | $44.6 \pm 0.3$ | $\mathbf{44.6 \pm 0.5}$ | $45.2 \pm 0.6$ | $\mathbf{51.6 \pm 0.9}$ |
| HALFCHEETAH-ME | $97.7 \pm 2.2$ | $\mathbf{101.1 \pm 0.4}$ | $99.4 \pm 2.5$ | $\mathbf{102.5 \pm 1.8}$ | $90.5 \pm 6.6$ | $\mathbf{93.9 \pm 2.9}$ | $81.1 \pm 6.0$ | $\mathbf{90.2 \pm 5.8}$ |
| HOPPER-M | $41.9 \pm 50.5$ | $\mathbf{103.0 \pm 0.8}$ | $45.7 \pm 41.0$ | $\mathbf{102.8 \pm 0.2}$ | $60.4 \pm 4.0$ | $\mathbf{61.0 \pm 2.6}$ | $65.0 \pm 6.1$ | $\mathbf{74.9 \pm 7.1}$ |
| HOPPER-MR | $92.5 \pm 18.1$ | $\mathbf{104.1 \pm 0.8}$ | $104.7 \pm 0.9$ | $\mathbf{104.6 \pm 0.4}$ | $61.2 \pm 20.5$ | $\mathbf{72.1 \pm 8.4}$ | $87.7 \pm 14.4$ | $\mathbf{100.3 \pm 3.2}$ |
| HOPPER-ME | $100.3 \pm 22.6$ | $\mathbf{110.7 \pm 0.6}$ | $110.9 \pm 0.2$ | $\mathbf{110.6 \pm 0.3}$ | $105.4 \pm 6.1$ | $\mathbf{104.8 \pm 2.8}$ | $93.9 \pm 14.3$ | $\mathbf{98.2 \pm 9.7}$ |
| WALKER2D-M | $69.9 \pm 35.2$ | $\mathbf{97.6 \pm 3.4}$ | $24.2 \pm 28.2$ | $\mathbf{92.3 \pm 1.3}$ | $82.7 \pm 5.5$ | $\mathbf{83.7 \pm 2.7}$ | $80.4 \pm 3.5$ | $\mathbf{84.5 \pm 1.0}$ |
| WALKER2D-MR | $91.6 \pm 2.7$ | $\mathbf{92.1 \pm 2.7}$ | $85.2 \pm 2.7$ | $\mathbf{86.9 \pm 3.1}$ | $82.1 \pm 2.5$ | $\mathbf{84.8 \pm 1.4}$ | $79.2 \pm 5.0$ | $\mathbf{94.4 \pm 2.5}$ |
| WALKER2D-ME | $90.6 \pm 45.0$ | $\mathbf{118.8 \pm 1.2}$ | $113.1 \pm 9.6$ | $\mathbf{116.4 \pm 1.5}$ | $110.2 \pm 0.5$ | $\mathbf{110.1 \pm 0.5}$ | $109.7 \pm 0.5$ | $\mathbf{113.0 \pm 0.5}$ |

Table 2: Average normalized scores of our methods and previous methods on the D4RL benchmark.

| DATASET | BC | DT | TD3BC | CQL | IQL | EDAC | DIFFUSION-QL | SAC-N-UNC +RD | TD3-N-UNC +RD |
|---|---|---|---|---|---|---|---|---|---|
| HALFCHEETAH-R | 2.2 | 2.2 | 11.0 | 31.3 | 13.7 | 28.4 | 22.0 | 25.4 | 31.0 |
| HOPPER-R | 3.7 | 5.4 | 8.4 | 5.3 | 8.4 | 25.3 | 18.3 | **31.6** | **31.7** |
| WALKER2D-R | 1.3 | 2.2 | 1.7 | 5.4 | 5.9 | 16.6 | 5.5 | 21.2 | 21.7 |
| HALFCHEETAH-M | 42.6 | 42.6 | 48.0 | 47.1 | 47.4 | 65.9 | 51.5 | 65.9 | **66.8** |
| HOPPER-M | 52.9 | 67.6 | 60.4 | 65.0 | 66.3 | 101.6 | 96.6 | 102.8 | **103.0** |
| WALKER2D-M | 75.3 | 74.0 | 82.7 | 80.4 | 78.3 | 92.5 | 87.3 | 92.3 | **97.6** |
| HALFCHEETAH-MR | 36.6 | 36.6 | 44.6 | 45.2 | 44.2 | 61.3 | 48.3 | **61.5** | 57.7 |
| HOPPER-MR | 18.1 | 82.7 | 61.2 | 87.7 | 94.7 | 101.0 | 102.0 | **104.6** | 104.1 |
| WALKER2D-MR | 26.0 | 66.6 | 82.1 | 79.2 | 73.9 | 87.1 | 98.0 | 86.9 | 92.1 |
| HALFCHEETAH-ME | 55.2 | 86.8 | 90.5 | 81.1 | 86.7 | 106.3 | 97.2 | 102.5 | 101.1 |
| HOPPER-ME | 52.5 | 107.6 | 105.4 | 93.9 | 91.5 | 110.7 | 112.3 | 110.6 | 110.7 |
| WALKER2D-ME | 107.5 | 108.1 | 110.2 | 109.7 | 109.6 | 114.7 | 111.2 | **116.4** | 118.8 |
| HALFCHEETAH-E | 91.8 | 87.7 | 96.7 | 97.3 | 94.9 | 106.8 | 96.3 | **108.8** | 103.1 |
| HOPPER-E | 107.7 | 94.2 | 107.8 | 106.5 | 108.8 | 110.1 | 102.6 | 109.8 | 108.8 |
| WALKER2D-E | 108.7 | 108.3 | 110.2 | 109.3 | 109.7 | 115.1 | 109.5 | 112.3 | 111.2 |
| MUJOCO TOTAL | 782.1 | 972.6 | 1020.9 | 1044.4 | 1034.0 | 1243.4 | 1158.6 | **1252.6** | 1259.4 |
| PEN-HUMAN | 25.8 | 73.9 | -1.9 | 35.2 | 71.5 | 52.1 | 75.7 | 61.1 | **77.9** |
| PEN-CLONED | 38.3 | 67.3 | 9.6 | 27.2 | 37.3 | 68.2 | 60.8 | 53.0 | 65.5 |
| ADROIT TOTAL | 64.1 | 141.2 | 7.7 | 62.4 | 108.8 | 120.3 | 136.5 | 114.1 | **143.4** |
| TOTAL | 846.2 | 1113.8 | 1028.6 | 1106.8 | 1142.8 | 1363.7 | 1295.1 | **1366.7** | **1402.8** |

To demonstrate the applicability of RD, we integrate RD into the training regime of TD3BC, CQL, TD3-N-Unc and SAC-N-Unc. As observed from Table 1, both TD3BC and CQL exhibit improved overall performance when RD is applied. It is noteworthy as the core idea of CQL to increase the Q value of the data in the dataset and diminish the Q value of the data generated by a mixed policy comprising the learning policy and random policy, which incorporates the representation distinction of state-action pairs sourced from different distributions. Therefore in practice, we apply the insights of RD to gradually transfer the original Q value restriction to the Q-value differentiation between the mixed policy and the designed OOD policy. As a result, CQL's performance is significantly improved. Moreover, the backbone algorithms themselves yield substantial advantages from the application of RD. Table 2 further reveals that backbone algorithms equipped with RD can achieve the best overall performance and attains state-of-the-art performance on several datasets. Additional evidence supporting the efficacy of RD in enhancing convergence speed and its potential to reduce the quantity of Q ensembles is provided in the appendix.

## 6.2 Comparison and discussion with techniques that inhibit overestimation

Table 3: Average normalized scores of TD3-N-Unc with RD or other variants, and DR3 and Layer Norm that inhibit potential overestimation

| DATASET | TD3-N-Unc | TD3-N-Unc +PDD | TD3-N-Unc +PDDDAW | TD3-N-Unc +RDD | TD3-N-Unc POD | TD3-N-Unc +DR3 | TD3-N-Unc +LayerNorm | TD3-N-Unc RD |
|---|---|---|---|---|---|---|---|---|
| HALFCHEETAH-M | $66.8 \pm 0.5$ | $66.1 \pm 0.9$ | $66.6 \pm 0.7$ | $65.9 \pm 0.8$ | $63.1 \pm 2.4$ | $64.4 \pm 1.7$ | $63.2 \pm 0.8$ | $\mathbf{66.8 \pm 1.2}$ |
| HOPPER-M | $41.9 \pm 50.5$ | $99.9 \pm 6.4$ | $100.7 \pm 6.2$ | $77.4 \pm 38.6$ | $82.8 \pm 40.5$ | $\mathbf{103.4 \pm 0.7}$ | $83.0 \pm 29.0$ | $103.0 \pm 0.8$ |
| WALKER2D-M | $69.9 \pm 35.2$ | $94.9 \pm 1.7$ | $66.7 \pm 38.1$ | $80.2 \pm 31.0$ | $94.2 \pm 4.7$ | $92.1 \pm 2.0$ | $68.5 \pm 20.7$ | $\mathbf{97.6 \pm 3.4}$ |
| HALFCHEETAH-E | $94.6 \pm 11.5$ | $96.3 \pm 13.2$ | $102.9 \pm 1.2$ | $99.9 \pm 4.5$ | $93.7 \pm 13.0$ | $100.0 \pm 3.7$ | $\mathbf{104.4 \pm 1.5}$ | $103.1 \pm 0.6$ |
| HOPPER-E | $110.9 \pm 0.5$ | $108.4 \pm 0.4$ | $108.4 \pm 0.4$ | $108.4 \pm 0.4$ | $108.2 \pm 0.7$ | $108.0 \pm 0.5$ | $88.4 \pm 42.8$ | $108.8 \pm 0.3$ |
| WALKER2D-E | $4.1 \pm 7.1$ | $67.4 \pm 47.2$ | $60.8 \pm 46.1$ | $28.5 \pm 40.8$ | $44.1 \pm 54.0$ | $109.9 \pm 0.4$ | $111.7 \pm 0.4$ | $\mathbf{111.2 \pm 0.7}$ |

We compare our method RD with the existing techniques that inhibit potential overestimation including DR3 [41] and Layer Norm [51]. In addition to the regularizer, all the other parameters are kept the same across different methods to ensure fairness. As shown in Table 3, RD is more helpful in helping improving the backbone algorithm than DR3 and Layer Norm.

RD and DR3 differ mainly from two perspectives. From the angle of the framework derived from, DR3 is derived from the theoretical characterizing implicit regularization in TD-Learning, which is a generalization of the implicit regularization from Supervised Learning [52, 53] to TD-Learning in RL setting. In contrast, RD is derived from the proposed Backup-Generalization framework. From the angle of regularization effect, DR3 is proposed to directly counter the implicit regularization of TD-Learning while RD is to suppress the generalization between in-sample data and OOD data. With some heuristics, DR3 regularizer arrives at a similar form, i.e., to minimize the NTK between consecutive state-action pairs in backup, to RD regularizer. Apparently, DR3 regularizer can be a special case of RD regularizer in terms of the definition of out-of-sample actions. The regularization effect of DR3 is similar to the Policy-Dataset Generalization Inhibition shown by Figure 2. Such a regularization can induce over-inhibition of generalization when the distribution of current policy overlaps with the offline dataset as illustrated. We also show in the following that DR3 and Policy-Dataset Generalization Inhibition (i.e., PDD in the following part) achieve similar empirical results.

## 6.3 Ablation Studies and In-depth Analysis

We design ablation experiments to elucidate the significance of RD's components. We compare RD against several variants including, including Policy-Dataset Distinction (PDD), Random-Dataset Distinction (RDD), Policy-OOD Distinction (POD), and Policy-Dataset Distinction with Dynamically Adjusted Weight (PDDDAW). PDD refers to the auxiliary representation loss calculated using data generated by the learning policy and that from the dataset. Likewise, RDD utilizes data generated by a random policy versus that from the dataset, while POD employs data produced by the learning policy against that by the OOD actor. As for PDDDAW, its PDD component is reweighted using the same heuristic $w$ adjustment method as RD, but the POD component is persistently weighted by zero. We draw a comparison between TD3-N-Unc + RD and these variants, as tabulated in Table 3. The results demonstrate that RD, fundamentally a dynamic amalgamation of PDD and POD, attains superior overall performance, underscoring the indispensability of each RD component.

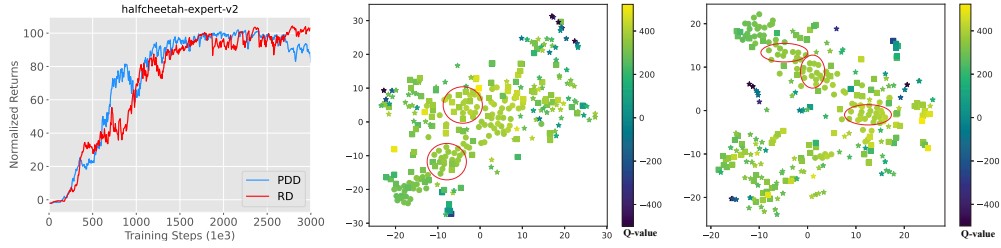

Figure 4: Performance using RD and PDD with same seed.

Figure 5: T-SNE visualization of representation via PDD.

Figure 6: T-SNE visualization of representation via RD.

It is worth noting that using PDD can help backbone algorithms achieve relatively satisfactory performance. However, as depicted in Fig 4, algorithm using PDD occasionally suffers performance degradation during the later stages of training, consequently impairing the overall efficacy of the approach. To elucidate this phenomenon, the final models trained via RD and PDD, as shown in Fig 4, are saved and the quality of the representations obtained by these models are compared. A total of 100 state-action pairs are sampled from the halfcheetah-expert, halfcheetah-medium, and halfcheetah-random datasets, respectively, and then consolidated. Using t-Distributed Stochastic Neighbor Embedding (t-SNE), the distribution of the representations of these 300 state-action pairs derived from both models is charted. Each sample is marked with distinctive symbols (stars for random data, squares for medium data, and circles for expert data) to differentiate samples from diverse datasets. Moreover, varying colors are assigned to individual samples corresponding to the Q-value estimated by the proficiently trained Q-value network acquired by RD, with brighter colors signifying higher Q-values. As depicted in Fig 5 and 6, the RD-derived representation exhibits superior differentiation between state-action pairs with high and low Q-values, whereas the PDD-derived representation frequently misclassifies expert and medium data.

To better demonstrate that the representations learned by RD are superior, we calculate which dataset the top five nearest samples to each sample belong to, and compute the mean over all samples from the same dataset in Table 4. For instance, the EXPERT–EXPERT metric of representation learned via RD indicates that 94% of the five nearest samples to the expert samples are also expert samples. We can conclude that representation learned via RD can better help samples from the same dataset to cluster closer. More results are provided in the appendix.

Table 4: The proportion of the five samples closest to each sample

|  | REP VIA PDD | REP VIA RD |
|---|---|---|
| EXPERT-EXPERT | 0.87 | 0.94 |
| EXPERT-MEDIUM | 0.12 | 0.06 |
| EXPERT-RANDOM | 0.01 | 0.00 |
| MEDIUM-EXPERT | 0.38 | 0.38 |
| MEDIUM-MEDIUM | 0.43 | 0.51 |
| MEDIUM-RANDOM | 0.19 | 0.11 |

## 7 Conclusions and Limitations

This paper analyze the important role of generalization in offline RL and presents a novel plug-in method, Representation Distinction (RD), to enhance the performance of offline RL algorithms by reining the generalization. By explicitly differentiating between in-sample and OOD state-action pairs, the generalization across data could be properly inhibited. Our extensive experiments demonstrate the efficacy of the RD method when applied to several offline RL algorithms, significantly enhancing their performance across various continuous control tasks on D4RL datasets. In conclusion, our work contributes a valuable perspective to the field of offline RL by focusing on reining generalization from the representation perspective and providing a flexible solution that can be applied to a variety of existing algorithms.

However, our work is not without limitations. For example, currently RD primarily focus on continuous control tasks, and its applicability to discrete control tasks is yet to be explored. In addition, the transition between the two phases of RD is a heuristic method, which could be improved by designing adaptive adjustment methods. Future research directions may include addressing the aforementioned limitation.

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
