## A  Hyperparameter Settings of RD

In this section, we describe details about hyperparameter setting of RD. There are several hyperparameters used in RD, including: $\alpha$, $\beta$ to control the OOD data generator to create data with lower Q-values compared to the data produced by current policy; $M$ to control the balance between $\mathcal{L}_1$ and $\mathcal{L}_2$; $\epsilon$ to balance $\mathcal{L}_{\text{RD}}$ and $\mathcal{L}_{\text{critic}}$ of any offline RL algorithm. Note that for all the datasets and algorithms used in the experiments, we set $\alpha = 0.6$, $\beta = 0.7$, $\epsilon = 0.1$. For $M$, we set it to 2e6 when applying RD to CQL. For other algorithms including TD3BC, SAC-N-Unc and TD3-N-Unc, $M$ is set to 1/10 of the total training steps.

In practice, for each in-sample state-action pair in $\mathcal{L}_1$ and state-action pair from the learning policy in $\mathcal{L}_2$, we sample 10 OOD state-action pairs and compute the loss respectively to differentiate more samples at one time to improve representation learning efficiency. For TD3-based algorithms, as the learning policy is determined, we additionally append a gaussian noise with scale of 0.01 on the sampled state-action pairs to obtain diverse samples.

In addition, we would like to emphasize that the OOD generator is not required to be meticulously trained, as we only expect the Q value of the generated OOD data to be appropriately lower than that of the data generated by the current policy. This is due to the fact that the OOD data is not employed to directly influence the policy's decision-making process. Meanwhile, $\mathcal{L}_{\text{RD}}$ is also not strictly required to minimized to zero as we only expect the state-action pairs from different sources could be distinguished instead of completely orthogonal in representation space. Therefore, the updating frequency of $\mathcal{L}_{\text{RD}}$ and $\mathcal{L}_{\text{OOD}}$, i.e., itv, is set to 10 as shown in Algorithm 1. To ensure fairness, algorithms employing RD are implemented using CORL repository [54].

## B  Backbone Algorithms

Here we introduce the two designed backbone algorithms, namely SAC-N-Unc and TD3-N-Unc, inspired by existing uncertainty-based methods [10, 43, 12, 44, 45]. By modifying the original SAC/TD3 algorithm to employ a critic ensemble of number N and incorporate an uncertainty regularization term within the policy update process, we derive these backbone algorithms. We clarify that the designing of backbone algorithms is not novel and they are only used to demonstrate the effectiveness of the proposed RD method in improving offline RL algorithms.

The optimization of the critic $Q_\phi$ and actor $\pi_\theta$ parameterized using the deep neural networks in SAC-N-Unc are as follows:

$$\mathcal{L}_{\text{critic}} = \min_{\phi_i} \frac{1}{|\mathcal{B}|} \mathbb{E}_{s,a,s'\sim\mathcal{B}} \left[ Q_{\phi_i}(s,a) - \left( r(s,a) + \gamma\mathbb{E}_{a'\sim\pi_\theta}\left[ \min_{j=1,2,...,N} Q_{\bar{\phi}_j}\left(s',a'\right) - \sigma\log\pi_\theta\left(a' \mid s'\right) \right] \right) \right]^2$$

$$\mathcal{L}_{\text{actor}} = \max_\theta \frac{1}{|\mathcal{B}|} \mathbb{E}_{s\sim\mathcal{B}} \left[ \min_{j=1,2,...,N} \mathbb{E}_{a\sim\pi_\theta}\left[ Q_{\phi_j}(s,a) - \sigma\log\pi_\theta(a \mid s) \right] - \eta U(s,a) \right]$$

$$(7)$$

where $\sigma$ is used to control the entropy. Similarly, by overloading $\mathcal{L}_{\text{critic}}, \mathcal{L}_{\text{actor}}, Q_\phi$ and $\pi_\theta$, we present the training of TD3-N-Unc below:

$$\mathcal{L}_{\text{critic}} = \min_{\phi_i} \frac{1}{|\mathcal{B}|} \mathbb{E}_{s,a,s'\sim\mathcal{B}} \left[ Q_{\phi_i}(s,a) - \left( r(s,a) + \gamma\mathbb{E}_{a'\sim\pi_\theta}\left[ \min_{j=1,2,...,N} Q_{\bar{\phi}_j}(s',a') \right] \right) \right]^2$$

$$\mathcal{L}_{\text{actor}} = \max_\theta \frac{1}{|\mathcal{B}|} \mathbb{E}_{s\sim\mathcal{B}} \left[ \min_{j=1,2,...,N} \mathbb{E}_{a\sim\pi_\theta}\left[ Q_{\phi_j}(s,a) \right] - \eta U(s,a) \right]$$

$$(8)$$

For both SAC-N-Unc and TD3-N-Unc, the uncertainty term is calculated as:

$$U(s, \pi_\theta(s)) = \sqrt{\frac{1}{|\mathcal{B}|} \sum_{j=1}^{N} \left( Q_{\phi_j}(s, \pi_\theta(s)) - \bar{Q}(s, \pi_\theta(s)) \right)^2}$$

$$(9)$$

where $Q_{\bar{\phi}}$ indicates the target network, $\bar{Q}$ represents the average Q-value of all Q networks, $\mathcal{B}$ means the sampled mini-batch data from dataset $\mathcal{D}$, $\eta$ is used to balance each component of actor loss and $i = 1, ..., N$. In practice, the state is also normalized similar to TD3BC [7]. For each task, the critic number $N$ and $\eta$ is set as following:

For other hyperparameters, SAC-N-Unc follows the settings in EDAC [10] and TD3-N-Unc follows the settings in TD3BC [7], respectively.

## C  Experimental Results

Here we provide evidence supporting the efficacy of RD in enhancing convergence speed and its potential to reduce the quantity of Q ensembles. As shown in Table 6, the performance at 1M and 3M steps of both backbone

Table 5: Hyperparameters used in the backbone algorithms

| DATASET | SAC-N-UNC | | TD3-N-UNC | |
|---|---|---|---|---|
| | $N$ | $\eta$ | $N$ | $\eta$ |
| HALFCHEETAH-R | 3 | 1 | 3 | 1 |
| HOPPER-R | 5 | 20 | 5 | 20 |
| WALKER2D-R | 20 | 1 | 20 | 1 |
| HALFCHEETAH-M | 3 | 1 | 3 | 1 |
| HOPPER-M | 10 | 10 | 3 | 10 |
| WALKER2D-M | 5 | 10 | 3 | 10 |
| HALFCHEETAH-MR | 3 | 1 | 3 | 1 |
| HOPPER-MR | 5 | 1 | 3 | 1 |
| WALKER2D-MR | 5 | 1 | 5 | 1 |
| HALFCHEETAH-ME | 5 | 10 | 5 | 10 |
| HOPPER-ME | 10 | 20 | 10 | 20 |
| WALKER2D-ME | 10 | 1 | 5 | 1 |
| HALFCHEETAH-E | 5 | 10 | 5 | 10 |
| HOPPER-E | 20 | 20 | 10 | 20 |
| WALKER2D-E | 10 | 10 | 10 | 10 |
| PEN-HUMAN | 30 | 5 | 30 | 10 |
| PEN-CLONED | 30 | 5 | 30 | 10 |

methods is improved by incorporating RD on a large portion of the datasets, demonstrating the effectiveness of RD in increasing convergence speed and final performance. Additionally, using RD with fewer Q ensembles can achieve similar or even better results than the backbone methods using more Q ensembles, indicating its potential in reducing computing resource consumption. The learning curves of the two backbone algorithms trained using RD are provided in Fig.8.

Fig.7 provides comparison between RD and several variants, including Policy-Dataset Distinction (PDD), Random-Dataset Distinction (RDD), Policy-OOD Distinction (POD), and Policy-Dataset Distinction with Dynamically Adjusted Weight (PDDDAW). For clear presentaion, the standard deviation is not plotted and the curves are smoothed. PDD refers to the auxiliary representation loss calculated using data generated by the learning policy and that from the dataset. Likewise, RDD utilizes data generated by a random policy versus that from the dataset, while POD employs data produced by the learning policy against that by the OOD actor. As for PDDDAW, its PDD component is reweighted using the same heuristic $w$ adjustment method as RD, but the POD component is persistently weighted by zero. We draw a comparison between TD3-N-Unc + RD and these variants, as tabulated in Table 3. The results demonstrate that RD, fundamentally a dynamic amalgamation of PDD and POD, attains superior overall performance, underscoring the indispensability of each RD component.

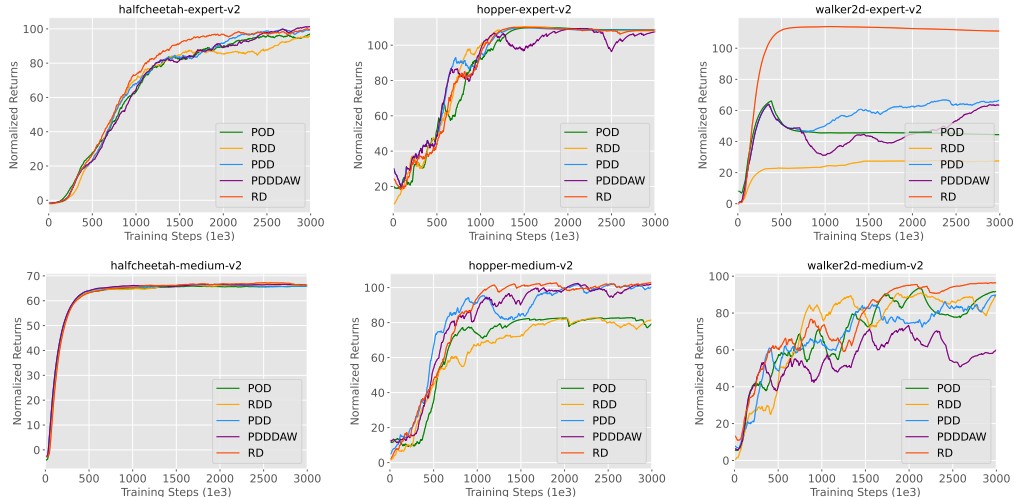

Figure 7: Learning curves using different representation distinction methods.

For a complementary, we further evaluate the similarity between different state-action pairs using two metrics: cosine similarity and L2 distance. We adopt a similar approach for both metrics. For instance, to obtain the EXPERT-EXPERT metric in Table 7, we compare each expert state-action pair to all other expert state-action pairs in the sampled batch, and select the top 5 closest pairs based on the metric. We then calculate the average cosine similarity or L2 distance across these five pairs and repeat this process for all expert state-action pairs. Finally,

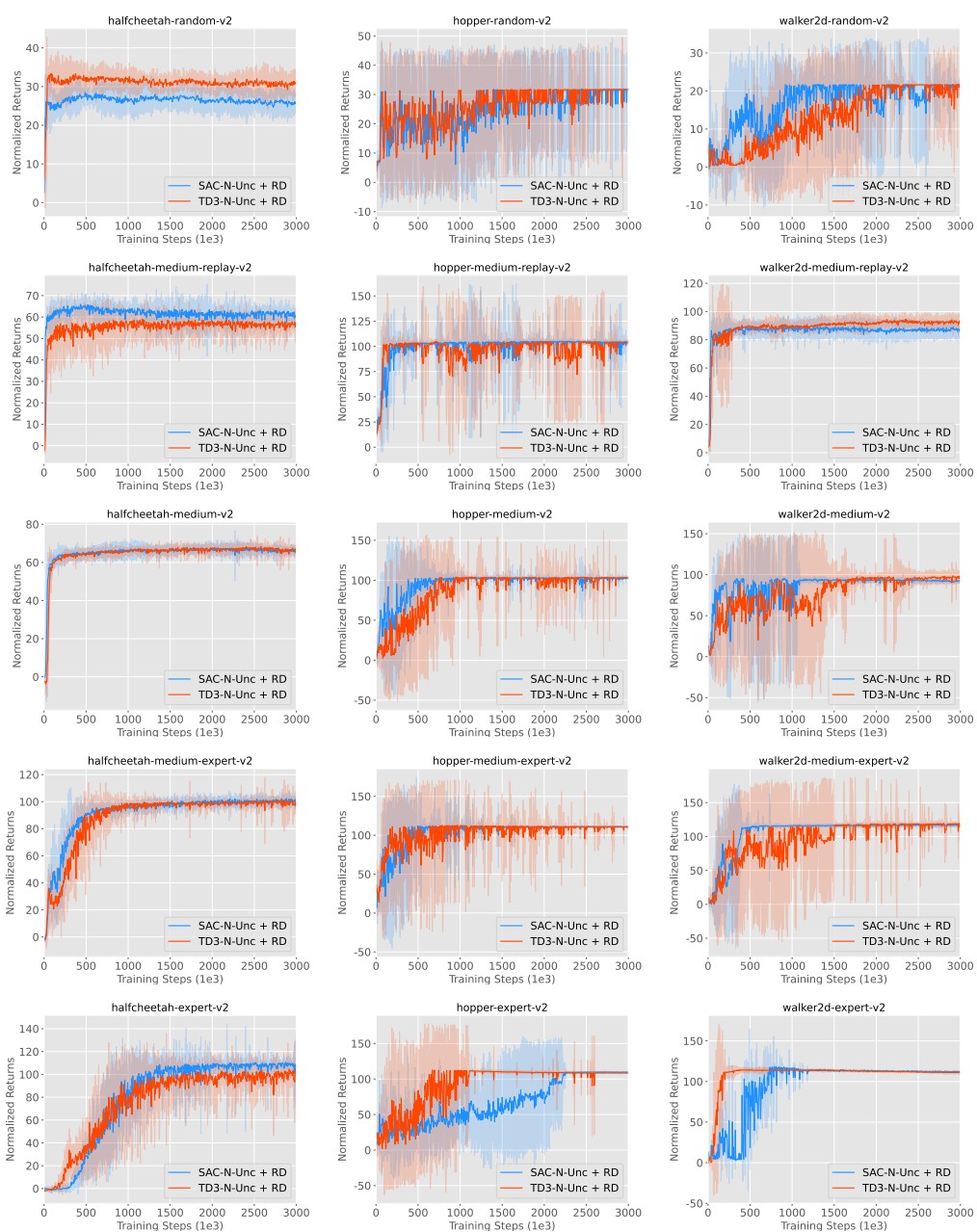

Figure 8: Learning curves of backbone algorithms using RD with 95% confidence interval.

Table 6: Empirical results of Representation Distinction (RD) for TD3-N-Unc and SAC-N-Unc on D4RL MuJoCo datasets. Same Q-num indicates the backbone algorithms have same critic number with that equipped with RD. More Q-Num the backbone algorithms have more critic number than that equipped with RD, and the specific critic number is twice as much as that equipped with RD. Mean scores and standard deviation errors across five seeds are reported.

| DATASET | METRICS | TD3-N-UNC (SAME Q-NUM) | **TD3-N-UNC (+ RD)** | TD3-N-UNC (MORE Q-NUM) | SAC-N-UNC (SAME Q-NUM) | **SAC-N-UNC (+ RD)** | SAC-N-UNC (MORE Q-NUM) |
|---|---|---|---|---|---|---|---|
| HA-R | SCORE (1M) | $33.0 \pm 2.2$ | $31.1 \pm 1.8$ | $33.1 \pm 1.2$ | $26.2 \pm 1.5$ | $26.8 \pm 2.1$ | $27.2 \pm 1.2$ |
| | SCORE (3M) | $31.6 \pm 1.9$ | $31.0 \pm 1.2$ | $31.4 \pm 1.1$ | $25.6 \pm 1.8$ | $25.4 \pm 0.6$ | $27.9 \pm 2.0$ |
| HA-MR | SCORE (1M) | $56.0 \pm 3.7$ | $56.5 \pm 3.1$ | $59.7 \pm 1.1$ | $61.6 \pm 2.6$ | $63.0 \pm 2.3$ | $63.1 \pm 5.2$ |
| | SCORE (3M) | $53.4 \pm 3.9$ | $57.7 \pm 0.9$ | $58.3 \pm 1.7$ | $53.2 \pm 5.4$ | $61.5 \pm 1.4$ | $62.8 \pm 1.4$ |
| HA-M | SCORE (1M) | $65.8 \pm 1.1$ | $65.3 \pm 1.4$ | $65.1 \pm 1.7$ | $65.2 \pm 1.9$ | $66.6 \pm 2.0$ | $68.0 \pm 1.6$ |
| | SCORE (3M) | $66.8 \pm 0.5$ | $66.8 \pm 1.2$ | $66.3 \pm 1.2$ | $69.5 \pm 1.0$ | $65.9 \pm 1.0$ | $69.8 \pm 1.9$ |
| HA-ME | SCORE (1M) | $85.7 \pm 4.8$ | $96.6 \pm 2.7$ | $96.5 \pm 1.3$ | $94.1 \pm 2.2$ | $95.9 \pm 2.5$ | $88.9 \pm 7.2$ |
| | SCORE (3M) | $97.7 \pm 2.2$ | $101.1 \pm 0.4$ | $90.0 \pm 6.0$ | $99.4 \pm 2.5$ | $102.5 \pm 1.8$ | $96.0 \pm 2.9$ |
| HA-E | SCORE (1M) | $33.0 \pm 15.1$ | $88.8 \pm 18.9$ | $83.3 \pm 17.2$ | $4.6 \pm 5.3$ | $70.1 \pm 16.0$ | $74.8 \pm 15.8$ |
| | SCORE (3M) | $94.6 \pm 11.5$ | $103.1 \pm 0.6$ | $100.3 \pm 1.8$ | $97.1 \pm 12.9$ | $108.8 \pm 1.3$ | $106.3 \pm 1.1$ |
| HO-R | SCORE (1M) | $25.4 \pm 11.9$ | $26.3 \pm 11.4$ | $26.9 \pm 9.0$ | $31.3 \pm 0.1$ | $22.4 \pm 11.0$ | $31.5 \pm 0.1$ |
| | SCORE (3M) | $31.8 \pm 0.1$ | $31.7 \pm 0.1$ | $31.7 \pm 0.1$ | $27.1 \pm 9.0$ | $31.6 \pm 0.2$ | $31.6 \pm 0.1$ |
| HO-MR | SCORE (1M) | $101.4 \pm 1.3$ | $87.7 \pm 34.9$ | $96.7 \pm 8.4$ | $103.4 \pm 0.5$ | $103.3 \pm 2.2$ | $102.4 \pm 0.2$ |
| | SCORE (3M) | $92.5 \pm 18.1$ | $104.1 \pm 0.8$ | $101.0 \pm 0.5$ | $104.7 \pm 0.9$ | $104.6 \pm 0.4$ | $102.3 \pm 0.5$ |
| HO-M | SCORE (1M) | $37.7 \pm 45.6$ | $97.4 \pm 13.3$ | $90.8 \pm 21.7$ | $40.5 \pm 38.5$ | $99.1 \pm 6.0$ | $95.6 \pm 7.5$ |
| | SCORE (3M) | $41.9 \pm 50.5$ | $103.0 \pm 0.8$ | $102.6 \pm 0.4$ | $45.7 \pm 41.0$ | $102.8 \pm 0.2$ | $94.3 \pm 9.4$ |
| HO-ME | SCORE (1M) | $70.0 \pm 29.3$ | $112.1 \pm 0.4$ | $109.9 \pm 4.7$ | $108.0 \pm 3.0$ | $110.4 \pm 0.4$ | $110.5 \pm 0.1$ |
| | SCORE (3M) | $100.3 \pm 22.6$ | $110.7 \pm 0.6$ | $93.7 \pm 22.7$ | $110.9 \pm 0.2$ | $110.6 \pm 0.3$ | $108.9 \pm 2.3$ |
| HO-E | SCORE (1M) | $52.0 \pm 55.7$ | $111.9 \pm 0.7$ | $61.4 \pm 49.8$ | $63.2 \pm 33.4$ | $46.6 \pm 18.8$ | $87.3 \pm 21.3$ |
| | SCORE (3M) | $110.9 \pm 0.5$ | $108.8 \pm 0.3$ | $109.1 \pm 0.4$ | $88.0 \pm 32.3$ | $109.8 \pm 0.3$ | $109.3 \pm 0.1$ |
| WA-R | SCORE (1M) | $3.4 \pm 5.6$ | $7.3 \pm 8.1$ | $21.5 \pm 0.2$ | $9.2 \pm 8.1$ | $18.3 \pm 2.7$ | $21.7 \pm 0.1$ |
| | SCORE (3M) | $15.5 \pm 6.4$ | $21.7 \pm 0.1$ | $21.2 \pm 0.8$ | $12.5 \pm 10.3$ | $21.2 \pm 0.7$ | $21.7 \pm 0.1$ |
| WA-MR | SCORE (1M) | $87.7 \pm 2.0$ | $90.3 \pm 2.0$ | $85.8 \pm 0.7$ | $84.5 \pm 3.4$ | $86.9 \pm 2.6$ | $83.3 \pm 1.8$ |
| | SCORE (3M) | $91.6 \pm 2.7$ | $92.1 \pm 2.7$ | $88.5 \pm 1.5$ | $85.2 \pm 2.7$ | $86.9 \pm 3.1$ | $85.5 \pm 1.3$ |
| WA-M | SCORE (1M) | $50.2 \pm 34.8$ | $85.0 \pm 14.5$ | $90.2 \pm 3.4$ | $26.2 \pm 20.3$ | $90.4 \pm 2.7$ | $85.5 \pm 1.0$ |
| | SCORE (3M) | $69.9 \pm 35.2$ | $97.6 \pm 3.4$ | $90.9 \pm 1.3$ | $24.2 \pm 28.2$ | $92.3 \pm 1.3$ | $74.5 \pm 6.0$ |
| WA-ME | SCORE (1M) | $85.9 \pm 43.2$ | $109.8 \pm 11.8$ | $114.9 \pm 0.9$ | $69.6 \pm 55.8$ | $116.0 \pm 1.0$ | $113.3 \pm 1.1$ |
| | SCORE (3M) | $90.6 \pm 45.0$ | $118.8 \pm 1.2$ | $116.6 \pm 0.5$ | $113.1 \pm 9.6$ | $116.4 \pm 1.5$ | $114.2 \pm 0.4$ |
| WA-E | SCORE (1M) | $4.0 \pm 2.1$ | $114.1 \pm 0.7$ | $108.9 \pm 0.4$ | $80.7 \pm 18.2$ | $116.5 \pm 0.3$ | $114.6 \pm 0.2$ |
| | SCORE (3M) | $4.1 \pm 7.1$ | $111.2 \pm 0.7$ | $107.1 \pm 0.2$ | $73.1 \pm 31.6$ | $112.3 \pm 0.1$ | $110.2 \pm 0.1$ |

we compute the mean of these average values to obtain the corresponding metric. Additionally, we compute the Q-value difference between each expert state-action pair and its nearest five expert state-action neighbors, and compute the mean of all Q-value differences across all expert state-action pairs to obtain the Q DIFF metric of EXPERT-EXPERT. As shown in Table 7, the representations of data from the same dataset learned via RD locate more closer in the representation space compared with that learned via PDD. The Q value difference between samples from different datasets is also more significant.

Table 7: Statistics of the representations learned using PDD and RD.

| OBJECT | METRIC | REP VIA PDD | REP VIA RD |
|---|---|---|---|
| EXPERT-EXPERT | COS SIM | 0.96 | 0.99 |
| | L2 DIS | 0.26 | 0.16 |
| | Q DIFF | -0.35 | -0.57 |
| EXPERT-MEDIUM | COS SIM | 0.90 | 0.94 |
| | L2 DIS | 0.71 | 0.45 |
| | Q DIFF | 17.40 | 20.50 |
| EXPERT-RANDOM | COS SIM | 0.72 | 0.82 |
| | L2 DIS | 2.26 | 1.41 |
| | Q DIFF | 65.13 | 83.85 |
| MEDIUM-EXPERT | COS SIM | 0.69 | 0.83 |
| | L2 DIS | 14.19 | 3.13 |
| | Q DIFF | -49.27 | -55.52 |
| MEDIUM-MEDIUM | COS SIM | 0.83 | 0.93 |
| | L2 DIS | 7.10 | 1.22 |
| | Q DIFF | -13.55 | -22.46 |
| MEDIUM-RANDOM | COS SIM | 0.72 | 0.81 |
| | L2 DIS | 10.29 | 2.51 |
| | Q DIFF | 37.11 | 37.97 |