# OpenReview forum: "Reining Generalization in Offline Reinforcement Learning via Representation Distinction"
_NeurIPS.cc/2023/Conference — NeurIPS 2023 poster_

### Official Review · Reviewer_XJcM · 2023-06-17

**Soundness:** 3 good
**Presentation:** 3 good
**Contribution:** 3 good
**Rating:** 6
**Confidence:** 3

**Summary:**

Authors consider overgeneralization in offline RL which is new and propose the modification which helps to solve the problem and  can be applied to any algorithm.

**Strengths:**

Proposed method boosted performance of different algorithms by a notable margin in most of the cases and it can be integrated in any algorithm while being simple. Good ablation study.

**Weaknesses:**

The problem epxeriments is that algorithms are tested using MuJoCo datasets and  2 pen datasets and  appeares not to be sufficient. For example, in CORL (https://arxiv.org/abs/2210.07105) benchmarks which you mention and  in original work (https://arxiv.org/abs/2110.01548), it is seen that EDAC performs great on the MuJoCo but fails on AntMaze (the recent update of CORL tested even more datasets and EDAC seems to fail on Adroit too). There are already some algorithms which are competitive or outperform EDAC on MuJoCo while working better on other datasets, see https://arxiv.org/abs/2206.02829 as an example. Some algorithms can performe not that good on MuJoCo but outperform others on different datasets, see concurrent work (https://arxiv.org/abs/2305.09836) or IQL in CORL.

I suggest to run more experiments using AntMaze or all of the Adroit datasets to verify that your approach helps. We can't conclude that it helps without checking more datasets. Wihout it I'm not sure that paper should be accepted.

**Questions:**

How does you modification affect training runtime? Is the change significant or not?

**Limitations:**

Limitations are not clear without additional benchmarking or runtime information.

---

> ### Author Rebuttal · Authors · 2023-08-09
>
> ### run more experiments using AntMaze or all of the Adroit datasets
>
> As AntMaze datasets are designed to investigate the stitching ability and the datasets can well cover the state-action space, our approach may not help significantly. Therefore, we perform experiments on Adroit datasets. We report average results across five seeds after training for 50 epochs on Adroit-expert datasets. On these datasets, RD significantly improve TD3-N-Unc's performance as shown in the following table. We also tried our best to tune algorithms on the remaining Adroit-cloned and Adroit-human datasets, however, it seems none of widely-used algorithms and backbone algorithms can obtain final models that have satisfying average performance, and our approach also failed to help improving the evaluation performance of final models. This phenomenon indicates RD cannot boost existing methods from the perspective of reining generalization on these datasets and more in-depth analysis of potential factors is required (e.g., limited data quantity), which is left for future works.
>
> | Dataset            | TD3-N-Unc       | TD3-N-Unc + RD  |
> | ------------------ | --------------- | --------------- |
> | pen-expert-v1      | 61.8 $\pm$ 26.1 | **84.7 $\pm$ 5.1**  |
> | door-expert-v1     | 11.1 $\pm$ 3.2  | **75.8 $\pm$ 29.4** |
> | relocate-expert-v1 | 1.9 $\pm$ 2.9   | **12.8 $\pm$ 9.0**  |
> | hammer-expert-v1   | 0.7 $\pm$ 0.9   | **20.9 $\pm$ 41.5** |
>
> Note that in addition to the auxiliary loss, all the other  parameters are kept the same across different methods to ensure fairness.
>
> ### How does you modification affect training runtime
>
> We counted the mean training time of one epoch (1000 gradient steps) of TD3-N-Unc and TD3-N-Unc + RD on all MuJoCo datasets,  the training time of TD3-N-Unc + RD is 1.3x that of TD3-N-Unc.  Although training time of one epoch is increased, we provide evidence in Table 6 (appendix C) of our paper, supporting the efficacy of RD in enhancing convergence speed and its potential to reduce the quantity of Q ensembles. As shown in Table 6, the performance at 1M and 3M steps of both backbone methods is improved by incorporating RD on a large portion of the datasets, demonstrating the effectiveness of RD in increasing convergence speed and final performance. Additionally, using RD with fewer Q ensembles can achieve similar or even better results than the backbone methods using more Q ensembles, indicating its potential in reducing computing resource consumption.

---

> > ### Comment · Reviewer_XJcM · 2023-08-11
> >
> > Thank you very much for answering my questions. Expert Adroit datasets are not that interesting but at least your approach improves results on them. I will increase the rating but decrease my confidence in it. Wishing you all the best with your publication.

---

> > > ### Author Response · Authors · 2023-08-17
> > >
> > > Thank you for the valuable advice.

---

### Official Review · Reviewer_yLhs · 2023-06-29

**Soundness:** 3 good
**Presentation:** 3 good
**Contribution:** 3 good
**Rating:** 6
**Confidence:** 4

**Summary:**

The authors investigated the effect of generalization in overestimation of critic values and devised an algorithm that can mitigate overgeneralization. The proposed method, Representation Distinction (RD), is orthogonal to most offline RL algorithms and thus be applied to most of them. Experimental results show that RD significantly enhances the performance of baseline algorithms.

**Strengths:**

### Originality
The paper introduces the Backup-Generalization framework to analyze the overestimation of critic values.
The authors devised a novel method of learning a suboptimal policy to sample OOD actions and use them in the later stage of training. They also proposed a simple and effective heuristic that allows a natural transition from PDD to POD as the training progresses.
### Quality
Most of the claims are technically sound. The authors have analyzed the performance of their method on popular offline RL benchmarks and conducted multiple interesting ablation studies.
### Clarity
The paper is overall well-written and easy to understand.
### Significance
Most works on offline RL focus on the backup step. This work instead points out the role of generalization in overestimation of critic values, which seems like a promising research direction. Also, the proposed method can be applied to most existing offline RL algorithms to improve their performance.


**Weaknesses:**

1. Line 315~316 is difficult to understand. A pseudocode of CQL+RD will be helpful.
2. A theoretical analysis of how RD differs from DR3 (Kumar et al. 2022) and why using OOD actions is better than just optimizing the DR3 auxiliary loss, $\mathbb{E}_{(s, a, s', a')\sim\mathcal{D}}\left[\Phi(s', a')^\intercal\Phi(s, a)\right]$, is missing from the paper. Adding this will improve the originality of the paper, as this work looks like a slight modification of DR3 at first glance. Also, the authors do not compare their algorithm with DR3 variants of CQL, TD3-BC, and SAC. Although I expect their performance to be on par with the PDD variants, I believe they should be included in the experiments section.
3. TD3-N-Unc and SAC-N-Unc use an ensemble of N critics. It is unfair to compare their performance with other baselines that only use one critic.

### References
Aviral Kumar, Rishabh Agarwal, Tengyu Ma, Aaron Courville, George Tucker, and Sergey Levine. DR3: Value-based deep reinforcement learning requires explicit regularization. In *International Conference on Learning Representations*, 2022.

**Questions:**

Table 3 reports the average normalized score of TD3-N-Unc on the Walker2D-Expert domain to be $4.1\pm 7.1$
and Table 1 reports the average normalized score of TD3-N-Unc on the Waler2D-Medium domain to be $69.9\pm 35.2$. Why does TD3-N-Unc perform a lot better on the medium dataset?

&nbsp;

### Minor suggestions

Eq. (4) It would be easier for the readers to follow if the authors explicitly mention that $\nabla_{\mathbf{w}}Q_\phi(s, a)=\Phi(s, a)$.

Line 315: gruadually $\to$ gradually

There are duplicate references: [33] and [41] are both Kumar et al. (2022).

**Limitations:**

The authors have addressed the limitations of their work.

---

> ### Author Rebuttal · Authors · 2023-08-09
>
> ### pseudocode of CQL+RD will be helpful.
>
> Here we provide the detailed illustration of modified CQL based on the insight of RD.  The core idea of CQL conservativeness is to increase the Q value of the data in the dataset $\mathcal{D}$ and diminish the Q value of the data generated by a mixed policy $\mu$ comprising the learning policy $\hat{\pi}\_\theta$ and random policy as following:
> $$
> \begin{aligned}
> \min\_Q \alpha\left(\mathbb{E}\_{\mathbf{s} \sim \mathcal{D}, \mathbf{a} \sim \mu(\mathbf{a} \mid \mathbf{s})}[Q(\mathbf{s}, \mathbf{a})]-\mathbb{E}\_{\mathbf{s} \sim \mathcal{D}, \mathbf{a} \sim \hat{\pi}\_\theta(\mathbf{a} \mid \mathbf{s})}[Q(\mathbf{s}, \mathbf{a})]\right) \\
> \end{aligned}
> $$
> This formulation incorporates the representation distinction of state-action pairs sourced from different distributions. Therefore in practice, we apply the insights of RD to gradually transfer the original Q value restriction in the above equation to the Q-value differentiation between the learning policy and the designed OOD policy in the below equation:
>
> $$
> \begin{aligned}
> \min\_Q \alpha\left(\mathbb{E}\_{\mathbf{s} \sim \mathcal{D}, \mathbf{a} \sim \hat{\pi}\_\theta(\mathbf{a} \mid \mathbf{s})}[Q(\mathbf{s}, \mathbf{a})]-\mathbb{E}\_{\mathbf{s} \sim \mathcal{D}, \mathbf{a} \sim \pi_\text{ood}(\mathbf{a} \mid \mathbf{s})}[Q(\mathbf{s}, \mathbf{a})]\right)
> \end{aligned}
> $$
>
> ### this work looks like a slight modification of DR3 at first glance
>
> RD and DR3 differ at both the theoretical framework from which they are derived and the regularization effect in terms of out-of-sample data.
>
> DR3 is derived from the theoretical characterizing implicit regularization in TD-Learning, which is a generalization of the implicit regularization effect from Supervised Learning as studied in Blanc et al. (2020) and Damian et al. (2021) to TD-Learning in RL setting. In contrast, RD is derived from the Backup-Generalization framework proposed by us.
>
> From the angle of regularization effect, DR3 is proposed to directly counter the implicit regularization of TD-Learning while RD is to suppress the generalization between in-sample data and out-of-sample data. With some heuristics, DR3 regularizer arrives at a similar form, i.e., to minimize the NTK between consecutive state-action pairs in backup $(s,a)$ and $(s^{\prime},a^{\prime})$ (where $a^{\prime} \sim \pi(\cdot|s^{\prime})$), to RD regularizer. Apparently, DR3 regularizer can be a special case of RD regularizer in terms of the definition of out-of-sample actions. The regularization effect of DR3 is similar to the Policy-Dataset Generalization Inhibition shown by Figure 2 in our paper. Such a regularization can induce over-inhibition of generalization when the distribution of current policy overlaps with the offline dataset as illustrated.
>
> We appreciate the reviewer's inspiring comment and we will add more analysis on the differences between RD and DR3 in our paper as suggested. For the empircal comparison between DR3 and RD, we provide additional experimental results as follows.
>
> ### comparisons with DR3
>
> According to the suggestion, we perform experiments against TD3-N-Unc with DR3 and with layer norm. Below table demonstrates the results on six datasets. Note that in addition to the auxiliary loss, all the other  parameters are kept the same across different methods to ensure fairness. Overall, RD is more helpful than DR3 and Layer Norm.
>
> | Dataset               | TD3-N-Unc + RD | TD3-N-Unc + DR3 | TD3-N-Unc + Layer Norm |
> | --------------------- | -------------- | --------------- | ---------------------- |
> | halfcheetah-medium-v2 | **66.8$\pm$1.2**   | 64.4 $\pm$ 1.7  | 63.2 $\pm$ 0.8         |
> | hopper-medium-v2      | **103.0$\pm$0.8**  | **103.4 $\pm$ 0.7** | 83.0 $\pm$ 29.0        |
> | walker2d-medium-v2    |  **97.6$\pm$3.4**   | 92.1 $\pm$ 2.0  | 68.5 $\pm$ 20.7        |
> | halfcheetah-expert-v2 | 103.1$\pm$0.6  | 100.0 $\pm$ 3.7 | **104.4 $\pm$ 1.5**        |
> | hopper-expert-v2      | **108.8$\pm$0.3**  | **108.0 $\pm$ 0.5** | 88.4 $\pm$ 42.8        |
> | walker2d-expert-v2    | **111.2$\pm$0.7** | 109.9 $\pm$ 0.4 | **111.7 $\pm$ 0.4**        |
>
>
>
> ###  unfair to compare performance with other baselines that only use one critic.
>
> Ensemble is a way to implement pessimism, which is concurrent to other types of pessimism such as policy constraint and value penalization. Beside, as noted in the appendix F of Gaon,et al (2021), increasing the number of Q-networks in CQL is of little help or even harmful. Thus aligning the number of Q-networks in each baseline algorithm may be not helpful. In addition, even for ensemble-based baseline EDAC, SAC-N-Unc+RD and TD3-N-Unc+RD can achieve better performance.
>
> ### Why does TD3-N-Unc perform a lot better on the medium dataset
>
> Table 3 is provided to elucidate the significance of RD’s components on different datasets. In order to achieve this, the hyperparameter settings of offline RL algorithm itself of each variant in Table 3 are exactly the same. This means that the results of the original TD3-N-Unc in Table 3 are not the optimal results it can achieve, so it is meaningless to compare the performance of the original TD3-N-Unc between different datasets. When we tune the hyperparameters of the original TD3-N-Unc, we can see in Table 6 of the appendix that the performance learned by TD3-N-Unc on the expert data set is better than that learned on the medium data.
>
> #### Reference
> * Guy Blanc, Neha Gupta, Gregory Valiant, and Paul Valiant. Implicit regularization for deep neural networks driven by an ornstein-uhlenbeck like process. In Conference on learning theory, pp.483–513. PMLR, 2020.
> * Alex Damian, Tengyu Ma, and Jason Lee. Label noise sgd provably prefers flat global minimizers. arXiv preprint arXiv:2106.06530, 2021.
> * An, Gaon, et al. "Uncertainty-based offline reinforcement learning with diversified q-ensemble." *Advances in neural information processing systems* 34 (2021): 7436-7447.

---

> > ### Comment · Reviewer_yLhs · 2023-08-12
> >
> > Thank you for the rebuttal. The DR3 results should definitely be included in the final version of the paper. I'll raise my score to 6, as most of my concerns have been resolved. By the way, does it mean that the CQL-RD does not have an NTK loss term?

---

> > > ### Author Response · Authors · 2023-08-17
> > >
> > > Thanks for your reply. Yes, CQL-RD does not have an explicit NTK loss term as the core idea of CQL is to increase the Q value of the data in the dataset and diminish the Q value of the data generated by a mixed policy comprising the learning policy and random policy, which incorporates the representation distinction of state-action pairs sourced from different distributions. Therefore, we apply the insights of RD to gruadually transfer the original Q value restriction to the Q-value differentiation between the mixed policy and the designed OOD policy.

---

### Official Review · Reviewer_S7qm · 2023-07-04

**Soundness:** 3 good
**Presentation:** 3 good
**Contribution:** 3 good
**Rating:** 6
**Confidence:** 4

**Summary:**

This paper explains the necessity of reining generalization in offline RL and proposed a new method Representation Distinction (RD) based on this principle. RD shows good performance on D4RL benchmarks.

**Strengths:**

- This paper presented a view called Backup-Generalization Cycle, which explains that the generalization issue should be considered in the training phase.
- This experiment shown in Figure 3 is interesting, which demonstrates overgeneralization is an important issue in offline RL and the method is able to mitigate it.
- By applying RD to existing algorithms, their performance is improved significantly.

**Weaknesses:**

- There is a gap between analysis and practical implementation. Therefore, it is difficult to identify whether the performance improvement comes from the overgeneralization issue.
- The algorithm introduces some hyperparameters which may make it a little difficult to tune.
- The difference between PDD and RD is not significant.

**Questions:**

- The experiment in Figure 3 is only conducted on halfcheetah environment, which is not very convincing as halfcheetah has some special features. Can the authors provide more results on other environments?
- In section 6.2, the authors only demonstrated the representation of  PDD and RD. While I'm curious about the representation of the original algorithm.

**Limitations:**

The limitation of this work is that it is limited to continuous control tasks.

---

> ### Author Rebuttal · Authors · 2023-08-09
>
> ### gap between analysis and practical implementation
>
> The conceptual illustration of reining generalization via kernel control in Section 4.3 including two stages: Policy-Dataset generalization inhibition and Policy-OOD generalization inhibition. In the practice algorithm design, we design loss  $\mathcal{L}_1$ and $\mathcal{L}_2$ for the two stages, respectively,  and design a simple heuristic approach to smoothly transit  from stage 1 to stage 2 by dynamically adjusting the weight of $\mathcal{L}_1$ and $\mathcal{L}_2$. Please let me know if you still find this correspondence is unclear.
>
> ###  hyperparameters which may make it a little difficult to tune
>
>  There are four hyperparameters used in RD, including: $\alpha$, $\beta$ to control the OOD data generator to create data with lower Q-values compared to the data produced by current policy; $M$ to control the balance between $\mathcal{L}\_{1}$ and  $\mathcal{L}\_{2}$; $\epsilon$ to balance $\mathcal{L}\_\text{RD}$ and  $\mathcal{L}\_\text{critic}$ of any offline RL algorithm. Note that for all the datasets and algorithms used in the experiments, we set $\alpha=0.6$, $\beta=0.7$, $\epsilon=0.1$. For $M$, we set it to 2e6 when applying RD to CQL. For other algorithms including TD3BC, SAC-N-Unc and TD3-N-Unc, $M$ is set to 1/10 of the total training steps.  The relatively general setting demonstrate the hyperparameters could be applied to different algorithms and datasets. Hyperparameters setting of RD are also provided in the appendix.
>
> ### difference between PDD and RD is not significant
>
> According to Table 3, the differences between PDD and RD on half of the datasets are relatively significant. On walker2d-expert dataset, the difference is quite large.  Although the differences between PDD and RD on some datasets are not clear, the differences between algorithm with PDD and the pure algorithm is quite significant, which demonstrates the effectiveness of PDD, which is an important component of RD a described in the main text of Section 6.2.
>
> ### more results on other environments in Figure 3
>
> We provide average results of SAC-2 (vanilla SAC) and SAC-2+RD of three seed after training 1M gradient steps on more dataset in the following table. It can be observed that RD is helpful in improving SAC-2 without explicit  pessimism.
>
> |  Dataset    | SAC-2 |  SAC-2 + RD |
> | -------- | ------- | ------- |
> | halfcheetah-random-v2  |     29.7       |   30.0   |
> | halfcheetah-medium-v2  |      38.2      |   68.5    |
> | halfcheetah-medium-replay-v2  |  0.8   |   49.2    |
> | halfcheetah-full-replay-v2 |   86.8     |   82.3    |
> | hopper-random-v2  |    9.9        |   15.2    |
> | hopper-medium-v2  |   0.8         |  2.1   |
> | hopper-medium-replay-v2  |  7.4   |   64.3    |
> | hopper-full-replay-v2 |   41.1     |    110.0   |
> | walker2d-random-v2  |     0.9       |   0.3    |
> | walker2d-medium-v2  |     -0.3       |   -0.2    |
> | walker2d-medium-replay-v2  |  -0.4   |  52.6     |
> | walker2d-full-replay-v2 |    27.9    |  97.3     |
>
>
> ### the representation of the original algorithm
>
> We expand Table 4 by adding corresponding statistics of the original algorithm on halfcheetah-expert-v2 in the following table. The performance of the original algorithm TD3-N-Unc of the same seed is 80.1, which is slightly lower than that of TD3-N-Unc with PDD ( 82.5).  It can be observed that both the representations learned by PDD and RD are better than that of the original algorithm.
>
> |  Dataset  |  Rep of ori | Rep via PDD |  Rep via RD |
> | -------- | ------- | ------- | ------- |
> | expert-expert | 0.83   | 0.87   | 0.94 |
> | expert-medium | 0.15   | 0.12   | 0.06  |
> | expert-random | 0.01   | 0.01   | 0.00  |
> | medium-expert | 0.38   | 0.38   | 0.38  |
> | medium-medium | 0.40   | 0.43   | 0.51 |
> | medium-random | 0.22   | 0.19   | 0.11   |

---

> > ### Comment · Reviewer_S7qm · 2023-08-15
> >
> > Thank you for the rebuttal. The authors' reply has addressed most of my concerns. Despite originating from different motivations, the similarity in methods between this work and previous studies downgrades the significance of this work. Therefore, I will maintain my current score.

---

> > > ### Author Response · Authors · 2023-08-17
> > >
> > > Thank you for the valuable advice. We will add the discussion of our method and existing methods in our paper.

---

### Official Review · Reviewer_xZbD · 2023-07-04

**Soundness:** 3 good
**Presentation:** 4 excellent
**Contribution:** 3 good
**Rating:** 6
**Confidence:** 3

**Summary:**

The paper proposes a flexible plug-in method called Representation Distinction (RD) to address the overgeneralization issue in offline Reinforcement Learning (RL) algorithms. The authors formalizes the process of generalization and investigates the potential to rein the generalization from the representation perspective to enhance offline RL, performing both Policy-Dataset and Policy OOD Generalization Inhibition with awareness of the two possible phases in the offline learning course. The proposed method explicitly differentiates between the representations of in-sample and out-of-distribution state-action pairs generated by the learning policy, which significantly improves the performance of the backbone and widely-used offline RL algorithms across various continuous control tasks on D4RL datasets.

**Strengths:**

+ **Clarity**: the paper is well-structured and clear to follow.
+ **Flexibility and Good Performance**: the authors propose a novel method called Representation Distinction (RD) to improve the performance of offline reinforcement learning algorithms, which explicitly differentiates between the representations of in-sample and out-of-distribution (OOD) state-action pairs generated by the learning policy, demonstrating the efficacy of the proposed approach by flexibly applying RD to specially-designed backbone algorithms and widely-used offline RL algorithms.
+ **Insight**: the paper provides an insightful, nuanced and systematic formalization of the process of generalization in offline RL and then investigates the prevalent overgeneralization issue in offline RL.

**Weaknesses:**

+ **Missing Citations or Clear Indications**: some recent papers also utilize NTK to analyze the genralization ability of offline RL on similar considerations with RD [1, 2, 3]. It is advisable to provide more comparisons and at least analyses on the way different NTK-relevant approaches address generalization issues in offline RL.
+ **Evaluation**: questions about experimental setups and results are listed in the next section.

[1] Kumar et al. DR3: Value-Based Deep Reinforcement Learning Requires Explicit Regularization. ICLR 2022.

[2] Ghasemipour et al. Why So Pessimistic? Estimating Uncertainties for Offline RL through Ensembles, and Why Their Independence Matters. NeurIPS 2022.

[3] Li et al. When data geometry meets deep function: Generalizing offline reinforcement learning. ICLR 2023.

**Questions:**

**Methodology**
+ It is recommended to include more comparisons and conduct thorough analyses on how RD and other existing NTK-relevant approaches address generalization issues in offline RL. This would enhance the comprehensiveness and depth of the discussion.
+ As far as I am concerned, RD via kernel control is doing value regularization implicitly on representation level. It would be beneficial to provide a clearer and more insightful explanation of the potential advantages of applying RD via kernel control instead of value regularization and policy constraint, particularly in the context of offline RL algorithms.

**Evaluation**
+ There appears to be an inconsistency between the results of SAC/TD3-N-Unc+RD in Table 1 and Table 2. It is important to address this discrepancy and provide clarification or additional information to ensure the accuracy and reliability of the reported findings.
+ Considering that well-established techniques that inhibit potential overestimation induced by extrapolation already applied into reinforcement learning, such as Lipchitz Regularization with Spectral Norm [4] and Layer Norm [5], are flexible and widely applicable, it would be worth discussing and experimenting with the possibility of leveraging these existing and simpler methods to effectively constrain and smooth the critic approximation, thereby preventing overgeneralization.
+ The comparison between curves obtained from a single seed in Figure 4 lacks experimental validity and may raise concerns about the reliability of the results. It is essential to address this limitation and consider conducting experiments with multiple seeds to improve the robustness and credibility of the presented findings.

[4] Gogianu et al.  Spectral normalisation for deep reinforcement learning: An optimisation perspective. ICML 2021.

[5] Ba et al. Layer normalization. Advances in NeurIPS 2016 Deep Learning Symposium, 2016.

**Limitations:**

 The authors have adequately addressed the limitations.

---

> ### Author Rebuttal · Authors · 2023-08-09
>
> ### analyses on how RD and other existing NTK-relevant approaches address generalization issues in offline RL
> See the global rebuttal parts.
>
>
> ### RD via kernel control is doing value regularization implicitly on representation level
>
> The core idea of RD is to encourage the learned Q-function to yield representations that are as orthogonal as possible between data from dataset and $\pi$ in the first stage and that between $\pi$ and $\pi_{ood}$  in the second stage.  Taking stage 1 for example,  in comparison to the value regularization on the q value **backup** process of $\pi$ (part of) in CQL, the value of $\pi$ using RD is regularized by cutting off the **generalization** from the learning on dataset. In extreme cases when $\left|\nabla_\phi Q_\phi(s, a)^{\top} \nabla_\phi Q_\phi(s, \pi(s))\right|=0$, the values of the unreliable actions of $\pi$ will be kept the same as the initialized ones. Although achieving similar results that the values of untrusted actions are low, the mechanism behind RD is completely different from the regularization of the Q value as shown in the Backup-Generalization Cycle in Figure 1.
>
> ### comparisons with DR3 and layer norm
> According to the suggestion, we perform experiments against TD3-N-Unc with DR3 and with layer norm. Below table demonstrates the results on six datasets. Note that in addition to the auxiliary loss, all the other  parameters are kept the same across different methods to ensure fairness. Overall, RD is more helpful than DR3 and Layer Norm.
>
> | Dataset               | TD3-N-Unc + RD | TD3-N-Unc + DR3 | TD3-N-Unc + Layer Norm |
> | --------------------- | -------------- | --------------- | ---------------------- |
> | halfcheetah-medium-v2 | **66.8$\pm$1.2**   | 64.4 $\pm$ 1.7  | 63.2 $\pm$ 0.8         |
> | hopper-medium-v2      | **103.0$\pm$0.8**  | **103.4 $\pm$ 0.7** | 83.0 $\pm$ 29.0        |
> | walker2d-medium-v2    |  **97.6$\pm$3.4**   | 92.1 $\pm$ 2.0  | 68.5 $\pm$ 20.7        |
> | halfcheetah-expert-v2 | 103.1$\pm$0.6  | 100.0 $\pm$ 3.7 | **104.4 $\pm$ 1.5**        |
> | hopper-expert-v2      | **108.8$\pm$0.3**  | **108.0 $\pm$ 0.5** | 88.4 $\pm$ 42.8        |
> | walker2d-expert-v2    | **111.2$\pm$0.7** | 109.9 $\pm$ 0.4 | **111.7 $\pm$ 0.4**        |
>
> ### single seed in Figure 4
>
> We use the result of one seed in Figure 4 to try to depict the occasionally happened performance degradation during the later stages of training in a clear manner.  The subsequent T-SNE visualization is also based on the model trained obtained in Figure 4.  Average results of several seeds could blur the result.  In the appendix, we provide the curves of five seeds, which could demonstrate the reliability of the results.
>
> ###  inconsistency between the results of SAC/TD3-N-Unc+RD in Table 1 and Table 2.
> Thanks for pointing out it. We have revised the results carefully and update the results in the draft.

---

> > ### Comment · Reviewer_xZbD · 2023-08-14
> >
> > Thanks for your rebuttal. The authors' reply has addressed most of my concerns and those additional comparisons with other methods definitely strengthen the paper's empirical claims so I will vote for a higher rate of acceptance.

---

> > > ### Author Response · Authors · 2023-08-17
> > >
> > > Thank you for the valuable advice. We will add the additional comparisons in our paper.

---

### Author Rebuttal · Authors · 2023-08-09


### analyses on how RD and other existing NTK-relevant approaches address generalization issues in offline RL
RD v.s. DR3 [1]

- RD and DR3 differ at both the theoretical framework from which they are derived and the regularization effect in terms of out-of-sample data. DR3 is derived from the theoretical characterizing implicit regularization in TD-Learning, which is a generalization of the implicit regularization effect from Supervised Learning as studied in [2] and [3] to TD-Learning in RL setting. In contrast, RD is derived from the Backup-Generalization framework proposed by us.

- From the angle of regularization effect, DR3 is proposed to directly counter the implicit regularization of TD-Learning while RD is to suppress the generalization between in-sample data and out-of-sample data. With some heuristics, DR3 regularizer arrives at a similar form, i.e., to minimize the NTK between consecutive state-action pairs in backup $(s,a)$ and $(s^{\prime},a^{\prime})$ (where $a^{\prime} \sim \pi(\cdot|s^{\prime})$), to RD regularizer. Apparently, DR3 regularizer can be a special case of RD regularizer in terms of the definition of out-of-sample actions. The regularization effect of DR3 is similar to the Policy-Dataset Generalization Inhibition shown by Figure 2 in our paper. Such a regularization can induce over-inhibition of generalization when the distribution of current policy overlaps with the offline dataset as illustrated.

RD v.s. MSG [4]
- MSG use NTK to characterize the difference of learning dynamics between Independent-taget ensemble and Shared-target ensemble for ensemble-based pessimistic offline RL methods, showing that commonly adopted Shared-target ensemble method could lead to a paradoxically optimistic estimate. The NTK analysis in MSG paper is not directly related to generalization issues in offline RL. Besides, MSG also has nothing to do with representation.
- One thing to notice is that MSG uses the general infinite-width NTK regime proposed by [5], while we follow the NTK notion especially in the context of RL proposed in [6]. Although the latter one is a derivative notion of the former one, they are often different in the settings and aims of analysis at least for RD and MSG here.

RD v.s. DOGE [7]

- DOGE uses NTK to characterize the difference of Q-function generalization for interpolated and extrapolated data, showing that Q-function generalizes better to interpolated state-action pairs. This then motivates the proposal of a new plug-in policy constraint that regularizes the learning policy to select actions in the approximated interpolated action space. Such a policy constraint can relieve the over-conservatism issue in prior policy constraint-based offline RL methods.
- In our paper, we use NTK to characterize the general generalization case within the Backup-Generalization framework. The major difference is that DOGE addresses the generalization issue by leveraging a less conservative policy constraint, while RD regularizes the generalization on the level of representation. To some degree, DOGE explicitly controls the policy space where  generalization issues are addressed; in contrast, RD can be viewed as realizing an implicit control of the policy space by the penultimate-layer representation. Interestingly, we think there is chance to make use the definition and approximation method of interpolated data and extrapolated data in DOGE paper to design new representation regularization schemes. We will consider this as a future direction.

We appreciate the reviewer's inspiring comment and we will add more analysis on the differences between RD and the related works above in our paper as suggested.
#### Reference
[1] Kumar et al. DR3: Value-Based Deep Reinforcement Learning Requires Explicit Regularization. ICLR 2022.

[2] Blanc et al. Implicit regularization for deep neural networks driven by an ornstein-uhlenbeck like process. In Conference on learning theory, pp.483–513. PMLR, 2020.

[3] Damian et al. Label noise sgd provably prefers flat global minimizers. arXiv preprint arXiv:2106.06530, 2021.

[4] Ghasemipour et al. Why So Pessimistic? Estimating Uncertainties for Offline RL through Ensembles, and Why Their Independence Matters. NeurIPS 2022.

[5] Jacot, A., Gabriel, F., and Hongler, C. Neural tangent kernel: Convergence and generalization in neural networks. arXiv preprint arXiv:1806.07572, 2018.

[6] Joshua Achiam, Ethan Knight, and Pieter Abbeel. Towards characterizing divergence in deep q-learning. arXiv preprint arXiv:1903.08894, 2019.

[7] Li et al. When data geometry meets deep function: Generalizing offline reinforcement learning. ICLR 2023.

---

### Decision · Program_Chairs · 2023-09-21

**Decision:**

Accept (poster)

**Comment:**

All reviewers are agreed that this submission should be accepted. After having gone through the paper myself, I agree.

I would recommend the authors improve their manuscript based on the suggestions given by the reviewers, and include the new experiments run during the rebuttals.